# SEA-SpeechBench: A Large-Scale Multitask Benchmark for Speech Understanding Across Southeast Asia

## Abstract

The rapid advancement of audio and multimodal large language models has un-locked transformative speech understanding capabilities, yet evaluation frame-works remain predominantly English-centric, leaving Southeast Asian (SEA) lan-guages critically underrepresented. We introduce SEA-SpeechBench, the first large-scale multitask benchmark that evaluates speech understanding in 11 SEA languages through more than 97,000 samples and 597 hours of curated audio data. Our benchmark comprises 9 diverse tasks across 3 categories: speech processing (automatic speech recognition, speech translation, spoken question answering), paralinguistic analysis (emotion, gender, age, speaker recognition), and tempo-ral understanding, a novel dimension featuring timestamped content queries and temporal localization within extended audio sequences up to 3 minutes. We imple-ment multilingual prompting in both native SEA languages and English to reflect user interactions with audio-language models. Evaluation of leading open-source and proprietary systems reveals marked performance gaps. Across all models, performance remains underwhelming on temporal reasoning, emotion recogni-tion, and speech translation, with most scores falling below 20. Prompting in low-resource languages such as Burmese, Lao, Tamil, and Khmer lag behind En-glish by over 5%. Our findings expose critical model limitations and underscore the need for inclusive model development. We will release datasets and the evalu-ation framework upon paper publication to facilitate reproducible benchmarking.

## 1 Introduction

Recent advancement in audio large language models (AudioLLMs) has led to transformative appli-cations in voice assistants, transcription, accessibility technologies, and multimodal reasoning (Wu et al., 2024; Gemini Team, 2025; Zhang et al., 2023). Despite these advances, research in speech understanding has been disproportionately concentrated on high-resource languages, particularly English and a small number of European or East Asian languages (Yang et al., 2021; Wang et al., 2021; Bu et al., 2017). While recent benchmarking efforts (Sakshi et al., 2024; Wang et al., 2024; Yang et al., 2024) have made significant strides in evaluating audio-language models across diverse tasks and modalities, they universally overlook Southeast Asian (SEA) languages, leaving an entire linguistic region underexplored despite representing over 650 million speakers worldwide.

Developing comprehensive benchmarks for SEA languages also presents unique technical chal-lenges. The region's speech landscape is characterized by extraordinary linguistic diversity, rich tonal and phonetic structures, and substantial resource disparities across languages: factors that create evaluation complexities absent from English-centric benchmarks. Many SEA languages op-erate in low-resource contexts with limited annotated data and sparse digital representation, making robust evaluation both methodologically challenging and critically important for equitable techno-logical development. While recent initiatives, such as MERaLiON (MERaLiON Team, 2024) which targets Singapore's multilingual context, Typhoon2-Audio (Pipatanakul et al., 2024) which focuses on Thai, and SeaLLMs-Audio (Liu et al., 2025) which extends capabilities to selected SEA lan-guages, have begun to build general-purpose speech-language models for the SEA region, these efforts remain limited in both scope and linguistic coverage. Crucially, they lack comprehensive evaluation frameworks necessary to systematically assess capabilities across the full spectrum of

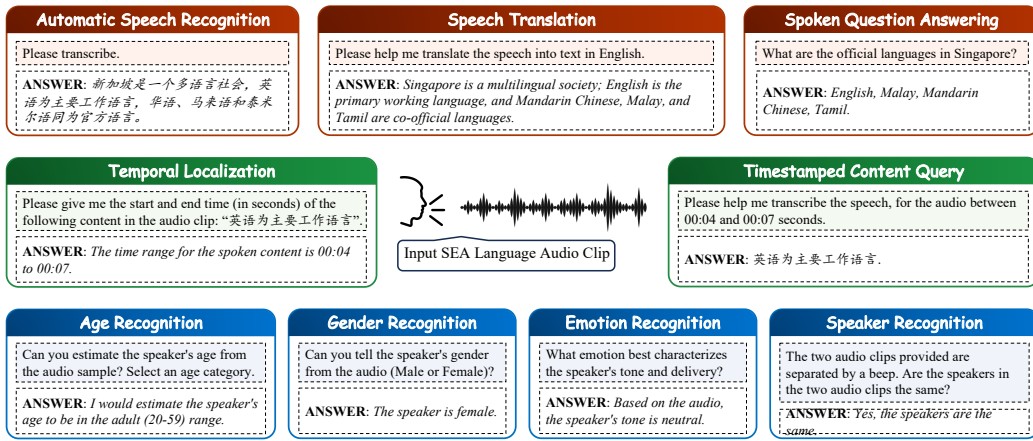

Figure 1: Overview of SEA-SPEECHBENCH task suite.The suite covers nine tasks in three categories. ■ **Speech processing**, ■ **Paralinguistic**, and ■ **Temporal understanding**.

Southeast Asian speech understanding tasks. Meanwhile, fragmented data collection efforts across research groups have yielded heterogeneous SEA datasets (Lovenia & et al., 2024; Pham et al., 2023; Bustamin et al., 2024; Magic Data Technology, 2025), but without unified frameworks for task definitions, prompting, and normalization, standardized comparison remains difficult.

In this work, we introduce the first ever comprehensive benchmark for speech understanding in Southeast Asian languages, designed to evaluate the capabilities of general-purpose speech-text LLMs. We focus on the official languages of Southeast Asian countries, as summarized in Table 1, to ensure broad regional coverage and practical relevance.

Table 1: Southeast Asian countries and their official language(s). Note: Several countries recognize additional regional or minority languages at sub-national levels; this table lists state-level official languages corresponding to the language codes used in our benchmark.

| Country | Official language(s) | ISO 639-1 Code(s) |
|---|---|---|
| Singapore | English, Malay, Mandarin Chinese, Tamil | en, ms, zh, ta |
| Malaysia | Malay (Bahasa Melayu) | ms |
| Indonesia | Indonesian (Bahasa Indonesia) | id |
| Philippines | Filipino / Tagalog, English | tl, en |
| Thailand | Thai | th |
| Cambodia | Khmer (Cambodian) | km |
| Lao PDR | Lao | lo |
| Myanmar | Burmese (Myanmar) | my |
| Vietnam | Vietnamese | vi |

Our benchmark encompasses 11 SEA languages with over 597 hours of audio data, curated from existing sources and synthesized into new tasks and datasets through systematic processing. The benchmark spans 9 diverse tasks across 3 broad categories: speech processing, paralinguistics, and temporal understanding. We introduce two temporal understanding tasks that evaluate models' capacity for temporal reasoning and localization within audio streams, addressing a previously unexplored dimension in audio LLM evaluation. These tasks test models' ability to navigate time-dependent information and extract content from specific temporal locations. The complete task suite is illustrated in Figure 1. To better reflect authentic usage scenarios, we evaluate each task using both English and native language text prompts.

**The main contributions of this paper are threefold.** First, we present SEA-SPEECHBENCH, the first ever large-scale multitask benchmark that systematically evaluates speech processing, paralinguistic analysis, and temporal understanding across Southeast Asian languages. It comprises a total of 99 evaluation sets with more than 97,000 audio samples. Second, we introduce temporal understanding tasks that assess models' ability to navigate and reason about time-dependent information in extended audio sequences. Finally, we conduct a comprehensive evaluation of both open-source

and proprietary models, offering critical insights into their strengths, limitations, and areas requiring further research.

## 2 RELATED WORKS

**Audio/Multimodal LLMs.** Recent language models have evolved to process spoken audio alongside text through diverse architectural approaches. Early work established key paradigms: alignment-based models (Tang et al., 2024; Zhang et al., 2023; Wu et al., 2023) connect speech encoders to LLMs via lightweight adaptors, while unified decoders (Rubenstein et al., 2023; Nguyen et al., 2024) share token spaces for joint speech-text modeling. Contemporary systems demonstrate varied innovations: Phi-4 (Microsoft, 2025) employs mixture-of-LoRAs for multimodality, Qwen2.5-Omni (Xu et al., 2025) advances temporal understanding through time-aligned position encoding, and Voxtral (Voxtral Team, 2025) handles extended recordings up to 40 minutes without separate ASR. Commercial systems like GPT-4o (OpenAI, 2024) and Gemini (Gemini Team, 2025) provide end-to-end audio understanding within unified frameworks. For Southeast Asian languages, MERaLiON (MERaLiON Team, 2024) targets Singapore's multilingual context while SeaLLMs-Audio (Liu et al., 2025) extends capabilities to five major SEA languages. However, these efforts remain limited in scope, overlooking lower-resource languages and lacking comprehensive evaluation frameworks for the region's full linguistic diversity.

**Audio/Multimodal LLM Benchmarks.** Audio-language evaluation has evolved from basic speech recognition to comprehensive multimodal assessment. While early efforts like SU-PERB (Huang et al., 2024) aggregated speech tasks universally, newer benchmarks (Wang et al., 2024; Yang et al., 2024) emphasize instruction-following across diverse audio types. Advanced benchmarks (Sakshi et al., 2024; Kumar et al., 2025) introduce complex reasoning with extended audio and multi-stream processing, while specialized evaluations target instruction-following (Gao et al., 2025) and domain-specific tasks (Ma et al., 2025). However, existing benchmarks remain predominantly English-centric with minimal Southeast Asian language coverage, creating a significant evaluation gap for low-resource linguistic contexts. This bias fundamentally limits understanding of model performance across the global linguistic landscape where inclusive deployment is most critically needed.

## 3 SEA-SPEECHBENCH: TASK, DATASET AND EVALUATION SUITE

### 3.1 TASK SUITE

SEA-SPEECHBENCH comprises 9 core tasks across 3 categories: *speech processing*, *paralinguistic analysis*, and *temporal understanding*. All tasks require models to produce textual responses given an audio input and a text query.

*Speech processing* covers three fundamental capabilities: Automatic Speech Recognition (ASR), Speech Translation (ST) from Southeast Asian languages to English, and Spoken Question Answering (SQA) based on SEA speech inputs.

*Paralinguistic analysis* examines vocal cues beyond linguistic content. It includes four tasks: Emotion Recognition (ER), which classifies emotional states; Gender Recognition (GR), which predicts gender from voice characteristics; Age Recognition (AgeR), which categorizes speakers as teens (10–19), adults (20–59), or seniors (60–100); and Speaker Recognition (SpkR), which determines whether two clips belong to the same speaker.

*Temporal understanding* introduces two novel tasks designed for extended audio, motivated by "skip to the content" and "what's said at this time" use cases. Timestamped Content Query (TCQ) requires extracting content within a specified interval $[t_s, t_e]$, testing temporal grounding and localized retrieval. Temporal Localization (TLoc) asks models to predict the exact time span $\hat{y} = [\hat{t}_s, \hat{t}_e]$ where queried information appears, evaluating boundary detection and alignment. Together, these tasks formalize time-referenced retrieval and support fine-grained navigation across recordings.

The first two categories use short clips ($\leq$ 30s) to align with current model input limits. As real-world audio applications increasingly involve longer recordings where users require temporal nav-

Table 2: Summary of curated datasets for SEA-SPEECHBENCH. For multilingual datasets, each language-specific sub-dataset is counted separately in #Datasets.

| Task | Languages | #Datasets | #Samples | Total L (h) | Min L (s) | Max L (s) |
|------|-----------|-----------|----------|-------------|-----------|-----------|
| **ASR** | en tl id km lo ms my ta vi zh th | 33 | 26,863 | 52.88 | 0.47 | 30.00 |
| **ST** | en tl id km lo ms my vi th | 9 | 7,189 | 26.46 | 3.06 | 30.00 |
| **SQA** | en zh id th vi | 7 | 5,462 | 40.57 | 20.00 | 30.00 |
| **ER** | zh id th en ta | 7 | 5,356 | 5.33 | 0.12 | 29.86 |
| **GR** | zh id th en ta vi km my | 16 | 13,599 | 22.02 | 0.12 | 29.90 |
| **AgeR** | zh th en ta vi | 5 | 4,608 | 6.55 | 0.58 | 20.78 |
| **SpkR** | zh th en ta vi my | 8 | 7,827 | 19.72 | 2.10 | 30.38 |
| **TCQ** | zh en th id vi | 7 | 13,145 | 211.98 | 20.00 | 180.00 |
| **TLoc** | zh en th id vi | 7 | 13,145 | 211.98 | 20.00 | 180.00 |
| **Total** | – | **99** | **97,194** | **597.49** | – | – |

igation, SEA-SPEECHBENCH introduces the temporal understanding tasks to assess model ability to perform reasoning and localization over extended sequences of up to 3 minutes.

## 3.2 DATA CURATION

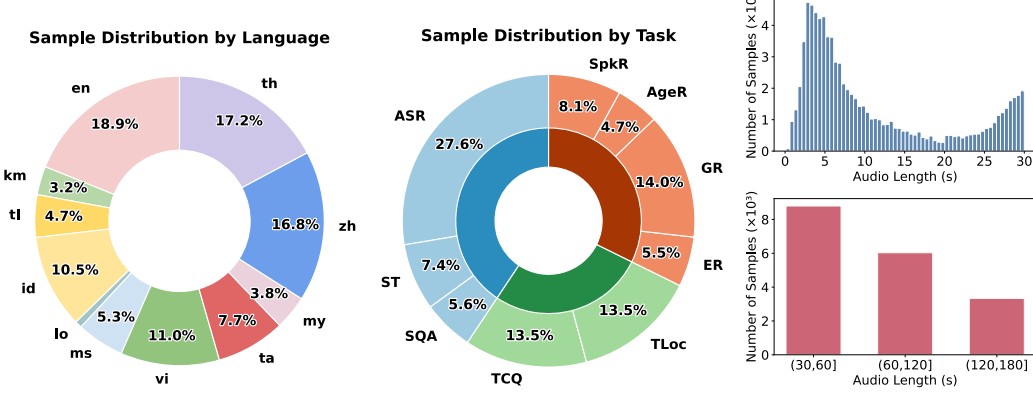

(a) Sample distribution by language. Colors match Table 2 language tags.

(b) Sample distribution by task.

(c) Audio duration distribution overview.

Figure 2: SEA-SPEECHBENCH composition: (a) language distribution by audio hours, (b) task distribution by sample count, and (c) audio duration distribution. (Upper: short audios ($\leq$ 30s) with fine-grained histogram. Lower: long audios ($>$ 30s) aggregated into three duration ranges.)

**Dataset Statistics:** As shown in Table 2, SEA-SPEECHBENCH is a comprehensive benchmark comprising of over 97,000 samples across 9 tasks and 11 Southeast Asian languages.

Traditional speech processing tasks span the broadest linguistic coverage with standard 30-second clips, while our novel temporal understanding tasks focus on 5 key languages with extended recordings up to 180 seconds. Detailed information for each source dataset is provided in Appendix A.

**Language and Task Distribution:** Figure 2 reveals the linguistic and task composition of our benchmark. The language distribution (Figure 2a) demonstrates substantial coverage across Southeast Asian languages, with English (18.9%), Thai (17.2%) and Chinese (16.8%) representing the largest segments, followed by Vietnamese (11.0%) and Indonesian (10.5%). We also include low-resource languages such as Khmer and Lao to ensure representation of the region's full linguistic spectrum. Figure 2b shows balanced coverage across evaluation task families.

**Audio Length Distribution**: We cover both short and long audios to enable more comprehensive and realistic evaluation. As shown in Figure 2c, our benchmark encompasses a broad distribution of utterances spanning 0–30 seconds, with natural concentration around sentence-length segments for traditional tasks. For temporal understanding evaluation, we systematically curate extended recordings through strategic segmentation of long-form datasets, creating stratified duration bins of 30–60s, 60–120s, and 120–180s, totaling over 10,000 samples for long audios.

We applied uniform data processing across all datasets and synthesized additional evaluation sets for tasks lacking suitable existing data. Processing and synthesis details are provided in Appendix B.

### 3.3 METRICS

Table 3 presents each task's output format and primary evaluation metric. For ASR and TCQ, where the expected outputs are text transcripts, we employ script-dependent error rates: Character Error Rate (CER) for languages without explicit word boundaries (Chinese (zh), Thai (th), Khmer (km), Lao (lo), and Burmese (my)), and Word Error Rate (WER) for the other space-delimited languages. For ST, we use the BLEU score to measure n-gram overlap between generated translations and reference texts. For TLoc, given a actual time span $y = [t_s, t_e]$ and a prediction $\hat{y} = [\hat{t}_s, \hat{t}_e]$, define the intersection $I = \left[ \min(\hat{t}_e, t_e) - \max(\hat{t}_s, t_s) \right]_+$ with $[x]_+ = \max(x, 0)$. Coverage and purity are $C = \frac{I}{t_e - t_s}$ and $P = \frac{I}{\hat{t}_e - \hat{t}_s}$, and the metric is $F1 = \frac{2CP}{C+P}$ for $C + P > 0$ (otherwise $F_1 = 0$).

For the other tasks, due to the inherent variability of free-form outputs, we employ an LLM-based judge for consistent evaluation, adapted based on the methodology proposed in Wang et al. (2024). For SQA, the judge assigns a 0–5 quality score; we report this as a percentage via a linear scaling $s_{SQA} = 20 \times \text{score}$, $\text{score} \in \mathbb{Z}_{[0,5]}$. For paralinguistic tasks (ER, GR, AgeR, SpkR), the judge canonicalizes outputs and renders binary correctness, reported as accuracy in percent. Detailed judging prompts are provided in the Appendix F.

To enable aggregation across heterogeneous metrics, we transform error rates $WER/CER$ into higher-is-better scores on the range $[0, 1]$ via the monotone reciprocal mapping: $s_{WER} = \frac{1}{1+WER}$, $s_{CER} = \frac{1}{1+CER}$. This transformation preserves relative ordering while ensuring well-defined scores even when excessive insertions yield $WER/CER > 1$. All remaining metrics in our evaluation framework are inherently higher-is-better, requiring no additional transformation. For presentation clarity, we report all metrics as percentages throughout our analysis. This aggregation facilitates direct comparison of overall model performance across tasks. Nonetheless, **aggregated scores should be viewed as a high-level summary**, and readers are encouraged to consult the individual metrics for detailed performance insights.

Table 3: Outputs and metrics by task. For ASR and TCQ, error-based metrics (WER/CER) are transformed into higher-is-better scores for comparability. [†] *Judge-provided metrics*: scores are produced by a LLM serving as an external judge.

| Task | Expected Output | Metric(s) |
|------|-----------------|-----------|
| **ASR** | Text transcript | $s_{WER}/s_{CER}$ |
| **ST** | Text translation | **BLEU** |
| **SQA** | Short textual answer | **Scaled judge score** $s_{SQA}$ [†] |
| **AgeR** | Age bin $\in$ {teens, adults, seniors} | |
| **ER** | Emotion label | |
| **GR** | Gender label $\in$ {male, female} | **Judge-based classification Acc** [†] |
| **SpkR** | Speaker identity match or mismatch | |
| **TCQ** | Text transcript at given time | $s_{WER}/s_{CER}$ |
| **TLoc** | Time span $[t_s, t_e]$ | $F1$ **score** |

## 4 EXPERIMENTS AND RESULTS

### 4.1 EXPERIMENTAL SETTING

**Evaluated Models.** We conduct extensive evaluation across several state-of-the-art open-source audio/multimodal LLMs with multiple size variants ranging from 2B to 10B parameters: MERaLiON-

2 (MERaLiON Team, 2024), SeaLLMs-audio (Liu et al., 2025), Phi-4-multimodal-instruct (Microsoft, 2025), Qwen2-Audio-Instruct (Chu et al., 2024), Qwen2.5-Omni (Xu et al., 2025), Gemma-3n-it (Gemma Team, 2025), Voxtral (Voxtral Team, 2025), and Kimi-audio (KimiTeam, 2025). All models are evaluated using their official released checkpoints with recommended inference configurations to ensure fair comparison. To establish comprehensive performance baselines, we also evaluate two leading commercial models: Gemini 2.0 Flash (Gemini Team, 2025) and GPT-4o (OpenAI, 2024). For GPT-4o, we employ the specialized Whisper-based transcription API for ASR tasks and the general audio understanding model for all other tasks, ensuring optimal performance.

**Model as Judge.** We leverage Gemma-3 27B Instruct (Gemma Team, 2025) as our evaluation judge to assign quality scores and correctness decisions for model outputs, chosen for its demonstrated reliability in canonicalizing free-form text responses across Southeast Asian languages. This approach ensures scalable and standardized evaluation across the diverse range of audio understanding tasks in our benchmark.

## 4.2 MAIN RESULTS

Table 4 presents our results grouped by task families. To allow fair comparison across models with different maximum input audio length capacities, this table focuses on clips shorter than 30 seconds. Results on >30 s audios for temporal reasoning tasks are discussed separately in Section 4.3.1. Comprehensive per-language results are provided in the Appendix G.

**Proprietary vs. Open-Source Models.** Proprietary models maintain a performance lead across tasks. Gemini 2.5 Flash achieves the highest overall scores across task categories, reflecting advantages in computational scale and proprietary training resources. Among open-source models, MERaLiON-2-10B emerges as the strongest performer, demonstrating speech processing capabilities that closely match GPT-4o's performance while achieving the second-best performance among open-source models on paralinguistic evaluations.

Table 4: Model performance across tasks under English (ENG) and SEA prompts. Category-average scores (%) are unweighted macro-averages over constituent tasks.

| Model | Size | Speech Processing | | | | | | | | Temporal Understanding | | | | | |
| | | ASR | | ST | | SQA | | Average | | TCQ (30s) | | TLoc (30s) | | Average | |
| | | ENG | SEA | ENG | SEA | ENG | SEA | ENG | SEA | ENG | SEA | ENG | SEA | ENG | SEA |
| Gemma-3n-it | 2B | 33.23 | 28.53 | 8.97 | 8.72 | 65.93 | 65.91 | 36.24 | 35.20 | 12.01 | 11.49 | 11.82 | 11.89 | 11.92 | 11.69 |
| Qwen2.5-omni | 3B | 59.70 | 47.31 | 7.60 | 6.27 | 67.50 | 56.64 | 44.94 | 40.1 | 10.27 | 14.34 | 30.49 | 25.45 | 20.38 | 19.90 |
| MERaLiON-2 | 3B | 70.92 | 69.93 | 7.56 | 7.42 | 60.08 | 62.47 | 46.19 | 46.61 | 17.32 | **17.83** | 18.82 | 17.47 | 18.07 | 17.65 |
| Voxtral | 3B | 41.32 | 40.82 | **19.98** | 18.15 | 74.33 | 76.03 | 45.21 | 45.00 | **17.42** | 17.68 | 17.85 | 16.47 | 17.63 | 17.08 |
| Gemma-3n-it | 4B | 44.04 | 30.46 | 10.98 | 13.58 | 71.85 | 74.51 | 42.29 | 39.52 | 12.89 | 12.70 | 13.25 | 13.98 | 13.07 | 13.34 |
| Phi-4 | 5.6B | 30.37 | 22.76 | 3.04 | 0.32 | 58.59 | 53.49 | 30.67 | 25.52 | 4.61 | 7.31 | 12.97 | 13.51 | 8.79 | 10.41 |
| SeaLLMs-Audio | 7B | 60.72 | 57.32 | 10.74 | 10.11 | 75.20 | 63.16 | 47.27 | 47.54 | 15.44 | 15.92 | 11.57 | 12.94 | 13.50 | 14.43 |
| Qwen2-Audio-it | 7B | 51.28 | 45.45 | 4.54 | 3.20 | 59.20 | 60.41 | 38.34 | 36.35 | 15.23 | 15.64 | 33.30 | 31.41 | 24.27 | 23.52 |
| Qwen2.5-omni | 7B | 44.68 | 47.96 | 7.91 | 8.06 | 61.50 | 66.03 | 38.03 | 40.69 | 15.41 | 14.30 | **35.74** | **33.08** | **25.57** | **23.69** |
| Kimi-Audio | 7B | 17.09 | 18.88 | 3.36 | 7.71 | 63.85 | 62.13 | 28.10 | 29.57 | 6.67 | 8.42 | 14.49 | 12.96 | 10.58 | 10.69 |
| MERaLiON-2 | 10B | **78.74** | **78.74** | 17.75 | **19.52** | 76.03 | 80.43 | **57.51** | **59.56** | 16.34 | 17.26 | 22.22 | 22.05 | 19.28 | 19.66 |
| Gemini 2.5 Flash | - | **87.72** | **87.72** | 16.86 | 18.89 | **84.51** | **87.02** | **63.03** | **64.54** | **29.33** | **28.79** | 11.66 | 13.59 | 20.49 | **21.19** |
| GPT-4o | - | 81.79 | 81.79 | **21.24** | **21.39** | 74.08 | 75.25 | 59.04 | 59.48 | 16.50 | 15.56 | **26.47** | **25.43** | **21.49** | 20.49 |

(a) Results on speech processing tasks (ASR, ST, SQA) and temporal understanding tasks (≤30 s) (TCQ, TLoc).

| Model | Size | Paralinguistic | | | | | | | | | |
| | | AgeR | | ER | | GR | | SpkR | | Average | |
| | | ENG | SEA | ENG | SEA | ENG | SEA | ENG | SEA | ENG | SEA |
| Gemma-3n-it | 2B | 65.91 | 38.76 | 12.21 | 13.17 | 27.71 | 14.65 | 42.35 | 39.32 | 37.05 | 26.47 |
| Qwen2.5-omni | 3B | 50.50 | 34.47 | 13.45 | 9.95 | 36.12 | 20.19 | 28.59 | 17.79 | 32.16 | 20.60 |
| MERaLiON-2 | 3B | 68.20 | 46.46 | 23.99 | 18.73 | 48.59 | 46.35 | 39.75 | 33.71 | 45.13 | 36.31 |
| Voxtral | 3B | 75.71 | 52.61 | 10.62 | 5.35 | 27.65 | 9.89 | 42.90 | 36.09 | 39.22 | 25.99 |
| Gemma-3n-it | 4B | **69.68** | **58.74** | 12.46 | 13.93 | 53.36 | 27.23 | 39.88 | 39.72 | 43.84 | 34.90 |
| Phi-4 | 5.6B | 43.45 | 41.98 | 20.87 | 9.20 | 44.21 | 32.60 | 23.27 | 27.48 | 32.95 | 27.82 |
| SeaLLMs-Audio | 7B | 63.16 | 11.64 | 12.34 | 9.17 | 54.97 | 43.66 | 53.59 | 34.37 | 46.01 | 24.71 |
| Qwen2-Audio-it | 7B | 20.21 | 23.13 | 24.47 | 19.36 | **92.25** | 66.31 | 49.85 | 36.56 | 46.70 | 36.34 |
| Qwen2.5-omni | 7B | 20.15 | 23.13 | 16.33 | 10.45 | 50.46 | 34.58 | 14.63 | 11.44 | 25.39 | 19.90 |
| Kimi-Audio | 7B | 47.91 | 48.42 | **36.86** | **42.55** | 90.97 | 79.23 | 54.16 | 48.16 | **57.48** | **54.59** |
| MERaLiON-2 | 10B | 66.24 | 57.31 | 18.73 | 20.34 | 59.93 | 46.28 | 53.25 | 46.12 | 49.54 | 42.51 |
| Gemini 2.5 Flash | – | 76.00 | **79.40** | 19.50 | 16.79 | **92.50** | **90.06** | **71.13** | **71.75** | **64.78** | **64.50** |
| GPT-4o | – | **76.60** | 61.20 | 17.00 | **19.50** | 46.13 | 31.63 | 10.50 | 8.38 | 37.56 | 30.18 |

(b) Results on paralinguistic tasks (AgeR, ER, GR, SpkR).

**Task Specialization.** Within specific task categories, several models exhibit particular strengths: Qwen2.5-omni-7B delivers the best open-source temporal localization performance, while Voxtral and SeaLLMs-Audio excel in spoken question answering tasks. Kimi-Audio demonstrates robust paralinguistic capabilities among open-source systems. Further details appear in Section 4.3.2.

**Failure Cases Analysis.** Temporal understanding tasks universally present the greatest challenge. Even leading commercial models achieve only approximately 20% scoring on TCQ and TLoc, substantially lower than traditional speech processing performance. Details of this task will be discussed separately in Section 4.3.1. Inadequate ST performance stems from two systematic failure modes. First, task confusion: many models treat ST as ASR, returning source-language transcripts instead of English translations. Second, target-language mismatch: outputs appear in non-English or code-switched text despite explicit English requirements in prompts. These failures persist across our standardized templates and bilingual prompting conditions, indicating weaknesses in cross-lingual semantic grounding. For ER, we observe negative emotion clustering: anger, disgust, fear, and sadness are frequently conflated, while low-arousal negative emotions are often collapsed to neutral. This reflects the small inter-class separations and fuzzy boundaries within the negative emotion family, compounded by limited training data for some languages, lead to elevated confusions. A striking anomaly emerges in GPT-4o's notably weak performance on speaker recognition (8.38-10.50%), contrasting sharply with its competitive results across other paralinguistic tasks. This underperformance will be discussed in Section 4.3.4.

**Robustness to Multilingual Prompts.** SEA language prompts consistently underperform relative to English prompts. The SEA–ENG gap is largest for paralinguistic tasks, plausibly because strict classification scoring penalizes parsing errors and refusals caused by weaker instruction-following under SEA prompts, compounded by non-Latin script tokenization issues. Gemini 2.5 Flash and Kimi-Audio prove most robust to prompt language variation, while SeaLLMs-Audio and Qwen2.5-omni-3B are the least robust. By contrast, best performing ASR models Gemini 2.5 Flash and MERaLiON-2-10B exhibit near-identical performance under both prompt languages, suggesting ASR task is heavily trained and largely prompt-agnostic. We provide more detailed analysis of cross-linguistic performance patterns and their underlying linguistic determinants in Section 4.3.3.

## 4.3 ANALYSIS AND INSIGHTS

### 4.3.1 TEMPORAL REASONING IN AUDIO LLMs

In this section, we provide detailed analysis of temporal understanding capabilities, stratifying performance across four duration bins: [0,30), [30,60), [60,120), [120,180) seconds as shown in Table 5a. Figure 5b presents coverage ($C$), purity ($P$), and $F_1$ of TLoc task across audio-duration ranges, using MERaLiON-2-10B as a case study.

Table 5: Temporal understanding performance by duration.

(a) TCQ and TLoc results (%) by duration, averaged across SEA and English prompts. A dash (–) indicates audio lengths for which the model is unable to perform inference.

(b) TLoc metrics for MERaLiON-2-10B.

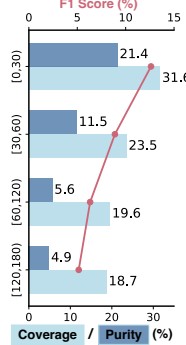

| Model | Size | TCQ | | | | TLoc | | | |
|---|---|---|---|---|---|---|---|---|---|
| | | 0–30 s | 30–60 s | 60–120 s | 120–180 s | 0–30 s | 30–60 s | 60–120 s | 120–180 s |
| SeaLLMs-Audio | 7B | 12.26 | - | - | - | 15.67 | - | - | - |
| Qwen2-Audio-it | 7B | 15.44 | - | - | - | 32.35 | - | - | - |
| Qwen2.5-omni | 3B | 11.97 | 10.82 | - | - | 27.97 | 17.02 | - | - |
| Gemma-3n-it | 2B | 11.75 | 11.82 | - | - | 11.85 | 8.45 | - | - |
| Gemma-3n-it | 4B | 12.79 | 12.04 | - | - | 13.61 | 8.95 | - | - |
| Phi-4 | 5.6B | 5.66 | 6.99 | 6.02 | - | 13.24 | 6.58 | 3.76 | - |
| MERaLiON-2 | 3B | **17.57** | 9.73 | 8.57 | - | 18.14 | 9.77 | 4.92 | - |
| Qwen2.5-omni | 7B | 14.84 | **13.08** | **10.42** | - | 34.41 | 20.54 | 11.02 | - |
| Kimi-Audio | 7B | 7.44 | 5.59 | 4.12 | 2.40 | 13.88 | 8.53 | 3.31 | 3.00 |
| Voxtral | 3B | 17.55 | 12.19 | 8.31 | 5.30 | 17.16 | 9.61 | 3.51 | 2.33 |
| MERaLiON-2 | 10B | 16.79 | 11.27 | 6.52 | **5.43** | 22.14 | 12.63 | 6.50 | **5.25** |
| Gemini 2.5 Flash | - | **29.05** | **27.17** | **13.85** | **19.83** | 12.62 | 8.89 | 6.33 | 5.15 |
| GPT-4o | - | 16.01 | 12.12 | 11.62 | 10.08 | 25.95 | 18.15 | 8.90 | 5.44 |

**Metrics Analysis.** First, we observe systematic over-coverage in temporal understanding across models. In TCQ, models frequently produce content that extends beyond the queried time window. In TLoc, as demonstrated in Figure 5b, coverage consistently exceeds purity across all durations, which is a pattern we observe in most evaluated models. This asymmetry indicates weak temporal

boundary localization and alignment, reflecting a recall-seeking strategy that favors longer spans, and thus higher coverage, at the cost of precision. These motivate finer-grained temporal grounding, boundary-aware training objectives, and decoding constraints that penalize span over-coverage.

**Constraints on Audio Length.** As shown in Table 5a, only a select subset of models sustains inference availability across all duration bins. Among open models, Voxtral, Kimi-Audio, and MERaLiON-2-10B demonstrate consistent availability, while commercial models handle the full range. Meanwhile, both TCQ and TLoc exhibit performance degradation with increasing duration: errors accumulate over longer contexts, manifesting as boundary drift and truncation, exposing current architectural limits in context window, frame compression, and long-range memory.

These findings underscore that temporal grounding represents an unresolved challenge in current audio-language architectures, with this deficiency becoming critically pronounced in long-form audio contexts where existing architectures prove inadequate for practical deployment.

### 4.3.2 Best-Performing Models by Language and Task

To highlight model strengths across tasks and languages, we plot a winner map that marks, for each task-language pair, the best-performing **open-sourced** model in Figure 3. Each cell shows the top model and color-codes model identity. MERaLiON-2-10B establishes clear dominance in speech processing tasks, consistently achieving top performance across multiple languages including Indonesian, Vietnamese, and Filipino. Paralinguistic task leadership proves more distributed, with Kimi-Audio and Qwen2-Audio-Instruct alternately excelling in different linguistic contexts. Temporal understanding tasks reveal limited model coverage and inconsistent performance patterns, suggests that model strengths are specialized rather than generalizable, with no single model demonstrating comprehensive temporal reasoning proficiency.

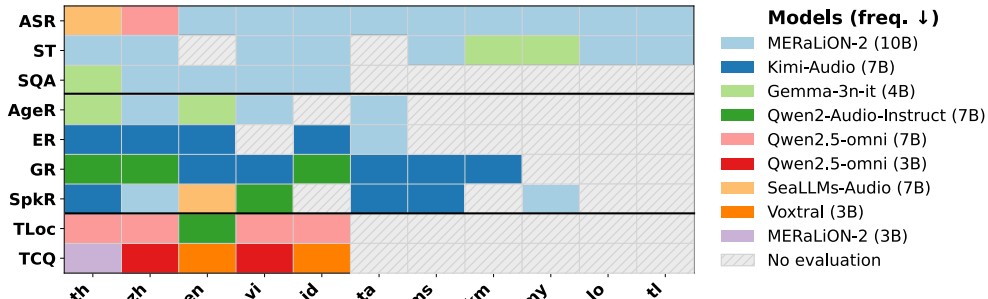

Figure 3: Winner map for open-sourced models across tasks and languages. Each cell marks the top model for a task–language pair; legend is ordered by overall win frequency.

### 4.3.3 Effect of Prompt Language

We systematically investigate how prompt language choice: English versus native SEA language, affects model performance across our benchmark.

Figure 4 demonstrates a cross-linguistic hierarchy in prompt sensitivity. We define a Prompt Advantage Score (PAS) to quantify this effect, with its detailed formulation provided in Appendix C. Higher PAS values indicate stronger local language prompt advantage, while negative scores suggest English prompt superiority for that particular language. Indonesian (id) emerges as the sole language showing consistent local prompt advantage (+0.3), while English (en) and Chinese (zh) exhibit near-neutral behavior (0.0, -0.2). The remaining SEA languages display increasing English preference across two distinct clusters: moderate disadvantages for Filipino, Vietnamese, Malay, Thai, and Myanmar (ranging from -1.7 to -5.3), and severe English advantages for Lao, Tamil, and Khmer (-8.7 to -20.8). This variation correlates strongly with orthographic and computational factors that influence instruction parsing effectiveness:

**Script Complexity and Tokenization.** Non-Latin scripts create fundamental computational barriers. Languages like Thai, Lao, and Khmer lack clear word boundaries and employ complex grapheme clusters that disrupt standard tokenization processes. Abugida systems such as Myanmar

and Tamil further complicate parsing through character-level ambiguities. These structural challenges impair instruction processing, while English prompts leverage well-established tokenization patterns that avoid such complications.

**Training Data Quality and Orthographic Consistency.** Corpus quality directly affects local prompt performance. Indonesian succeeds due to abundant Latin-script training data with consistent orthographic standards and imperative constructions, supporting reliable instruction following. In contrast, lower-resource languages suffer from limited, inconsistent training corpora marked by orthographic variations and dialectal diversity, making local prompts less reliable than standardized English alternatives.

These results expose a critical deployment gap: when users issue prompts in Southeast Asian languages, which is the natural interaction mode for regional populations, performance degrades substantially compared to English-prompted evaluation. This asymmetry underscores the urgent need for multilingual instruction-tuning that aligns with authentic user interaction patterns throughout model development and evaluation.

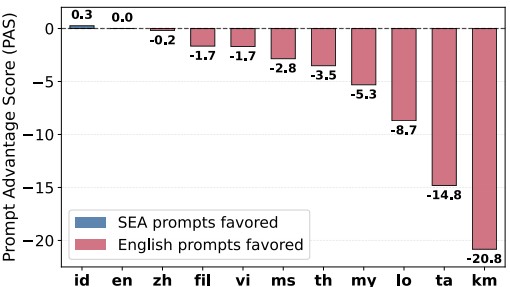

Figure 4: Cross-linguistic prompt sensitivity measured by Prompt Advantage Score (PAS).

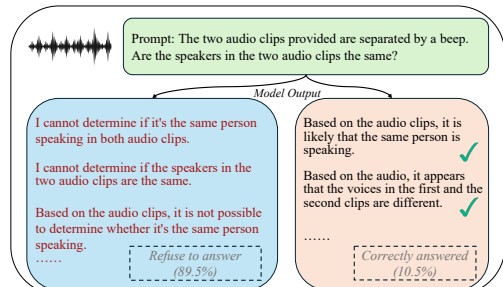

Figure 5: Speaker recognition failure examples in GPT-4o responses.

### 4.3.4 REFUSAL BEHAVIOR: CAUSES AND PREVALENCE

From Table 4, GPT-4o attains only 10.50% accuracy on speaker recognition. Inspecting the error breakdown shows that these "errors" are refusals rather than wrong predictions. As illustrated in Figure 5, GPT-4o refuses to answer 89.5% of the queries. When the model does answer for 10.5% of the queries, it is consistently correct (non-refusal accuracy = 100%). This pattern emerges on our self-constructed SpkR dataset, which is likely out-of-distribution for GPT-4o, suggesting limited task generalization. Concurrently, the model may adopt a conservative, uncertainty-aware strategy rather than making overconfident predictions. By contrast, Qwen2.5-Omni-7B also attains low SpkR performance with frequent refusals, but its accepted responses contain nontrivial mistakes, pointing to weaker calibration and label grounding rather than abstention alone.

## 5 CONCLUSION

We present SEA-SPEECHBENCH, the first comprehensive benchmark for evaluating speech understanding across 11 Southeast Asian languages, comprising 97,000+ samples across 9 tasks in speech processing, paralinguistics, and temporal understanding. Our standardized framework enables reproducible, cross-linguistic comparisons through unified normalization and bilingual prompting.

Evaluation of leading commercial and open-source systems exposes systematic weaknesses: performance collapses on long audio (temporal brittleness), English prompts consistently outperform native languages (linguistic inequity), and tasks such as temporal reasoning, emotion recognition, and speech translation remain far below usability thresholds. These findings underscore persistent scalability and generalization gaps. By surfacing these limitations, SEA-SPEECHBENCH seeks to establishe a rigorous baseline for developing temporally robust, linguistically inclusive, and practically deployable speech technologies for Southeast Asia's diverse communities.

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
