# A    DATASET CATALOG

Table 6: Overview of datasets: Dataset statistics, including language coverage, task types, basic audio properties, and licensing information.

| | Languages | Tasks | Total L (hr) | Min L (s) | Max L (s) | License |
|---|---|---|---|---|---|---|
| **Commonvoice** (Ardila et al., 2020) | zh, vi, th, id, ta, en | GR, AGE, ASR | 22.66 | 0.58 | 20.78 | MPL 2.0 |
| **FLEURS** (Conneau et al., 2022) | zh, vi, th, my, ms, lo, km, id, tl | ASR, ST, GR | 57.79 | 3.06 | 30.00 | CC-BY 4.0 |
| **OpenSLR** (Sodimana et al., 2018) (He et al., 2020; Oo et al., 2020) | ta, my, km | GR, ASR | 6.33 | 1.71 | 17.15 | CC-BY-SA 4.0 |
| **Bloom-Speech** (Leong et al., 2022) | tl, my, en | ASR | 0.84 | 0.47 | 29.46 | CC BY-NC 4.0 |
| **Thai Elderly Speech** (Lovenia & et al., 2024) | th | ASR, GR, SpkR | 5.66 | 2.04 | 27.06 | CC-BY-SA 4.0 |
| **LOTUS** (Chotimongkol et al., 2009) | th | ASR | 1.22 | 1.84 | 29.78 | CC-BY-NC-SA 4.0 |
| **THAI SER** (Lovenia & et al., 2024) | th | SpkR, GR, ER | 6.22 | 0.60 | 29.86 | CC-BY-SA 4.0 |
| **SG Streets** (Khassanov et al., 2019) | en | TLoc, TCQ, SQA, ASR, GR | 6.55 | 0.80 | 59.97 | NA |
| **VietMed** (Pham et al., 2023) | vi | ASR | 1.74 | 2.00 | 12.00 | MIT |
| **VoxVietnam-O** (Vu et al., 2025) | vi | SpkR | 2.43 | 1.42 | 29.68 | CC |
| **Bud500** (Pham et al., 2024) | vi | ASR | 0.71 | 1.01 | 5.48 | Apache-2.0 |
| **Vietnam-Celeb** (Pham et al., 2023) | vi | GR | 2.14 | 0.84 | 29.66 | CC-BY-4.0 |
| **ASR-SMalDuSC** (Lovenia & et al., 2024) | ms | ASR, SpkR, GR | 8.51 | 2.64 | 29.22 | CC BY-NC-ND 4.0 |
| **ASR-MALCSC** (Magic Data Technology, 2025) | ms | ASR | 0.74 | 0.61 | 18.16 | CC BY-NC-ND 4.0 |
| **IndoWaveSentiment** (Bustamin et al., 2024) | id | GR, ER | 0.52 | 3.00 | 3.80 | CC-BY-4.0 |
| **EmoTa** (Thevakumar et al., 2025) | ta | SpkR, GR, ER | 2.90 | 1.06 | 9.79 | EACL |
| **SFDUSC** (Magic Data Technology, 2023) | tl | SpkR, GR, ASR | 4.86 | 2.03 | 21.49 | CC BY-NC-ND 4.0 |
| **YODAS2** (Li et al., 2023) | en, id, zh, th, vi | TLoc, TCQ, SQA | 493.12 | 20.01 | 180.00 | CC-BY 3.0 |
| **ESD** (Zhou et al., 2022) | en, zh | ASR, ER, SpkR | 6.56 | 1.41 | 10.79 | MIT |
| **M3ED** (Zhao et al., 2022) | zh | ER, GR | 0.83 | 0.12 | 7.04 | CC BY-NC-ND 4.0 |
| **MIG** (Myanmar Innovative Group, 2025) | my | ASR, SpkR | 1.50 | 0.80 | 21.39 | NA |
| **TEC** (Thanushs25, 2024) | ta | ER | 0.68 | 2.69 | 29.86 | NA |
| **ASR-SgpCCSC** (Magic Data Technology) | zh | ASR | 43.76 | 0.88 | 180.00 | CC BY-NC-ND 4.0 |

# B    DATA PROCESSING AND SYNTHESIS

We provide additional details on data processing and the construction of new datasets from source materials.

## B.1    DATA PROCESSING

**Audio and Prompt Standardization:** All audio was resampled to 16 kHz for consistency and model compatibility. We provide parallel prompts in native SEA languages and English to reflect realistic usage patterns, constructing standardized templates where needed. Prompt examples are provided in the Appendix E.

**Dataset Sampling:** 1. We adopted official test splits when available or sampled 1,000 instances while preserving identifiers to prevent contamination. 2. For our evaluation set, we further sampled 1,000 instances per dataset using class-balanced sampling for classification tasks, ensuring computational feasibility while maintaining representativeness.

This standardized processing transforms diverse datasets into a unified, reproducible benchmark supporting consistent evaluation across models.

## B.2    NEW EVALUATION DATA CONSTRUCTION

**SpkR (Speaker Recognition)**: We constructed datasets by sampling and concatenating pairs of audio segments, separated by a short beep, drawn either from the same speaker or from different speakers, to create controlled instances for the speaker verification task. Refer to Table 6 for the datasets used to synthesize the speaker recognition task.

**SQA (Spoken Question Answering)**: We carefully filtered the YODAS2 dataset (Xinjian et al., 2023) using CTC forced alignment scores between the audio clips and the provided transcriptions with a log-probability threshold of -1, exact language match between the audio and text transcription, as well as profanity and content filtering. Based on the cleaned transcriptions, we generated question–answer pairs using GPT-4.1 (OpenAI, 2024).

**TCQ (Timestamped Content Query)**: We filtered the YODAS2 dataset following the same procedure used for SQA. We then identified utterances whose CTC log-probability exceeded -0.1,

recording their start and end timestamps. The task was formulated such that models are required to transcribe utterances occurring between the specified timestamps.

**TLoc (Temporal Localization)**: We adopted the same preprocessing procedure as in TCQ. The task was formulated such that models are required to predict the start and end timestamps corresponding to specified utterances.

In all cases, we applied the same standardized audio preprocessing, sample selection, and prompt construction pipeline, ensuring these synthesized datasets are consistent and compatible with the broader benchmark.

## C    DEFINITION OF PROMPT ADVANTAGE SCORE (PAS)

We quantify prompt-language effects with a scale-invariant score capturing both *magnitude* and *direction*. Let $s_{\text{SEA}}$ and $s_{\text{EN}}$ denote task scores for SEA language prompt and English prompt, respectively. For each model $M$, task $T$, and language $L$, define the symmetric relative difference

$$d_{M,T,L} = \frac{\left| s_{\text{SEA}} - s_{\text{EN}} \right|}{\frac{|s_{\text{SEA}}| + |s_{\text{EN}}|}{2} + \varepsilon}, \tag{1}$$

and aggregate across models within task by the median $\tilde{d}_{T,L} = \text{median}_M \, d_{M,T,L}$. Directionality is encoded by the model-wise win rate of local prompts $w_{T,L} = \mathbb{P}_M\big(s_{\text{SEA}} > s_{\text{EN}}\big)$. The task-level Prompt Advantage Score combines direction and effect size:

$$\text{PAS}_{T,L} = \big(2(w_{T,L} - 0.5)\big) \cdot \tilde{d}_{T,L}, \tag{2}$$

so $\text{PAS}_{T,L} > 0$ favors local prompts and $\text{PAS}_{T,L} < 0$ favors English. The language-level score averages over the tasks covered by $L$, denoted $\mathcal{T}(L)$:

$$\text{PAS}_L = \frac{1}{|\mathcal{T}(L)|} \sum_{T \in \mathcal{T}(L)} \text{PAS}_{T,L}. \tag{3}$$

By construction, $|\text{PAS}_L|$ measures the strength of prompt sensitivity while $\text{sign}(\text{PAS}_L)$ indicates the preferred prompt language.

## D    POSTPROCESSING OF EVALUATION

To ensure fair and consistent evaluation across Southeast Asian languages, we design a unified text normalization pipeline comprising three sequential stages:

**Canonical Form Conversion.** Transcripts undergo initial standardization through NFC Unicode normalization and systematic lowercasing to eliminate encoding inconsistencies and case variations. Configurable punctuation filtering preserves linguistically meaningful characters (hyphens, apostrophes) while removing extraneous symbols that could introduce evaluation noise.

**Content Filtering and Lexical Standardization.** Non-linguistic artifacts including speaker tags, bracketed annotations, and conversational fillers are systematically removed. Digital content undergoes targeted normalization, and English contractions receive consistent expansion. For character-based writing systems, e.g., Chinese and Thai, inter-character spacing is inserted to enable reliable tokenization.

**Language-Specific Enhancement and Validation.** We collaborated with native speakers across all SEA languages to validate and refine normalization heuristics.

# E    EVALUATION PROMPT EXAMPLES

Figure 6: Overview of prompt examples: illustrative prompts provided for ASR, ST, SQA, AgeR, and ER tasks across multiple languages.

| Tasks | Prompt Language | Input Example | Answer Example |
|---|---|---|---|
| Automatic Speech Recognition | ENG | **Text:** Please help me transcribe the speech into text in English.🔊: [English Speech] | 🖥: Activity. Draw a picture of you playing a sport. |
| | | **Text:** Listen to the speech and provide the text version in Malay.🔊: [Malay Speech] | 🖥: Angin kencang, hujan batu, pemendakan berlebihan dan kebakaran hutan adalah bentuk dan kesan cuaca yang teruk. Begitu juga ribut petir, tornado, semburan air dan siklon. |
| | SEA | **Text:** Sila bantu saya transkripsikan ucapan ini ke dalam bentuk teks dalam Bahasa Melayu. 🔊: [Malay Speech] | 🖥: Angin kencang, hujan batu, pemendakan berlebihan dan kebakaran hutan adalah bentuk dan kesan cuaca yang teruk. Begitu juga ribut petir, tornado, semburan air dan siklon. |
| | | **Text:** กรุณาช่วยถอดเสียงพูดนี้เป็นข้อความในภาษาไทยด้วยค่ะ 🔊: [Thai Speech] | 🖥: มันเกิดอุบัติเหตุที่ทำงาน ทำให้อูฐเกือบฆ่าคนเฝ้าประตู |
| Speech Translation | ENG | **Text:** Please help me translate the speech into text in English 🔊: [Malay Speech] | 🖥: The television reports that the white smoke seen is from plants. |
| | | **Text:** Recognize the verbal content in the speech and translate it into text in English. 🔊: [Thai Speech] | 🖥: The smaller the Rossby number, the less movement the star will have in relation to the reversal of the Earth's magnetic poles. |
| | SEA | **Text:** အင်္ဂလိပ် သို့ စကားကို ပြောချက်ကို စာသားအဖြစ် ဘာသာပြန်ပေးရန် ကျေးဇူးပြုပါ။ 🔊: [Burmese Speech] | 🖥: Strong winds, heavy rain and sleet, and lightning are forms and effects of severe weather such as thunderstorms, tornadoes, waterspouts, and cyclones. |
| | | **Text:** Vui lòng giúp tôi chuyển lời nói thành văn bản bằng Tiếng Anh 🔊: [Vietnamese Speech] | 🖥: In his writing on the presidential speech, Oliver Sacks points out how people who are unable to understand the speech due to brain damage can still accurately judge its truthfulness. |
| Spoken Question Answering | ENG | **Text:** How many housing units were scheduled to be built in Bedok under the mentioned plan? 🔊: [English Speech] | 🖥: Sixteen thousand five hundred housing units. |
| | | **Text:** What can the playback tab be used for? 🔊: [English Speech] | 🖥: It can be used to play videos or adjust effects. |
| | SEA | **Text:** Untuk apa sendok diambil dalam proses mengambil air sirih merah? 🔊: [Indonesian Speech] | 🖥: Sendok diambil agar bisa merendam semua sirih merah. |
| | | **Text:** Vì sao người nói cảm thấy buồn đau và phải chôn giấu nỗi buồn đó? 🔊: [Vietnamese Speech] | 🖥: Vì đi qua nhiều mối tình đổ vỡ mà không ai ở lại, khiến người nói cảm thấy buồn đau và phải chôn giấu nỗi buồn đó. |
| Age Recognition | ENG | **Text:** Can you guess the speaker's age based purely on their voice characteristics? Choose among these age categories: teens (10-19), adult (20-59), or senior (60-100). 🔊: [English Speech] | : adult (20-59) |
| | | **Text:** Can you estimate the speaker's age from the audio sample? Select an age category: teens (10-19), adult (20-59), or senior (60-100). 🔊: [Thai Speech] | 🖥: Based on the audio sample, the speaker is likely in the adult (20-50) age category. |
| | SEA | **Text:** பேச்சை அடிப்படையாகக் கொண்டு, பேசும் நபருக்கு ஏற்ற சராசரி வயது வரம்பை வழங்க முடியுமா? தயவுசெய்து ஒரு வயது பிரிவை தேர்ந்தெடுக்கவும்: இளையோர் (10-19), பெரியோர் (20-59), அல்லது மூத்தோர் (60-100). 🔊: [Tamil Speech] | 🖥: பெரியோர் (20-59) |
| | | **Text:** จากเสียงของผู้พูด คุณคิดว่าผู้พูดอยู่ในช่วงอายุหรือกลุ่มอายุใด? กรุณาเลือกหนึ่งกลุ่มอายุจาก: วัยรุ่น (10-19), ผู้ใหญ่ (20-59), ผู้สูงอายุ (60-100) 🔊: [Thai Speech] | 🖥: ผู้ใหญ่ (20-59) |
| Emotion Recognition | ENG | **Text:** Based on the speaker's speech patterns, what do you think they are feeling? 🔊: [English Speech] | 🖥: The speaker's speech suggesting they might be feeling neutral. |
| | | **Text:** Can you identify or describe any emotions or feelings expressed by the speaker? Answer only using one sentence. Do not explain. 🔊: [Tamil Speech] | : happy |
| | SEA | **Text:** คุณสามารถระบุหรืออธิบายอารมณ์หรือความรู้สึกที่ผู้พูดแสดงออกมาได้หรือไม่? กรุณาตอบด้วยประโยคเดียวเท่านั้น ห้ามอธิบายเพิ่มเติม 🔊: [Thai Speech] | 🖥: ผู้พูดแสดงความรู้สึกหงุดหงิด ไม่พอใจ และอารมณ์ฉุนเฉียวอย่างรุนแรง |
| | | **Text:** Apa interpretasi Anda terhadap emosi yang ditunjukkan dari petunjuk emosional yang ada dalam audio? Berikan jawaban satu kalimat tentang emosi tersebut. Jangan berikan alasan. 🔊: [Indonesian Speech] | 🖥: Emosi yang ditunjukkan adalah kepuasan. |

Figure 7: Overview of prompt examples: illustrative prompts provided for GR, SpkR, TCQ, and TLoc tasks across multiple languages.

| Tasks | Prompt Language | Input Example | Answer Example |
|---|---|---|---|
| **Gender Recognition** | ENG | **Text:** Can you identify the speaker's gender based on the audio (Male or Female)? 🔊: **[English Speech]** | 🍰: male. |
| | | **Text:** From the audio, can you guess the speaker's gender (Male or Female)? 🔊: **[Thai Speech]** | 🍰: female. |
| | SEA | **Text:** ஆடியோவை அடிப்படையாகக் கொண்டு பேசும் நபரின் பாலினத்தை (ஆண் அல்லது பெண்) அடையாளம் காண முடியுமா? 🔊: **[Tamil Speech]** | 🍰: ஆண். |
| | | **Text:** Dapatkah Anda membedakan jenis kelamin pembicara berdasarkan audio ini (Laki-laki atau Perempuan)? 🔊: **[Indonesian Speech]** | 🍰: Laki-laki. |
| **Speaker Recognition** | ENG | **Text:** A beep sound is placed between the two audio segments. Is there a match between the identity of the speakers in the two recordings? 🔊: **[English Speech]** | 🍰: Yes, the speakers are the same. |
| | | **Text:** A short beep marks the break between the two audio clips. Do you think both clips feature the same speaker? 🔊: **[Burmese Speech]** | 🍰: No, the speakers are different. |
| | SEA | **Text:** Bunyi beep diletakkan di antara dua segmen audio. Adakah terdapat padanan identiti penutur dalam kedua-dua rakaman ini? 🔊: **[Malay Speech]** | 🍰: Tidak, pembesar suara berbeza. |
| | | **Text:** Hai đoạn âm thanh được cung cấp được ngăn cách bởi một tiếng bíp. Liệu người nói trong hai đoạn âm thanh này có phải là cùng một người không? 🔊: **[Vietnamese Speech]** | 🍰: Không, loa thì khác nhau. |
| **TCQ** | ENG | **Text:** Please help me transcribe the speech into text in English, for the audio between 21.96 and 25.54 seconds. 🔊: **[English Speech]** | 🍰: The first phase of the eastern line which included aljunied. |
| | | **Text:** Process the speech and provide the text output in Thai for the duration 6.07 to 9.6 seconds. 🔊: **[Thai Speech]** | 🍰: การซ้อมที่ดีที่คุณต้องซ้อมเสมือนว่าคุณกำลังพูด |
| | SEA | **Text:** Nghe đoạn phát biểu và cung cấp phiên bản văn bản bằng Tiếng Việt, bao gồm phần từ 12.03 đến 16.38 giây. 🔊: **[Vietnamese Speech]** | 🍰: Tính thuế cho từng cá nhân hoàn toàn tự động để nộp về từng quốc gia sở tại. |
| | | **Text:** Ubah masukan ucapan menjadi transkripsi teks dalam Bahasa Indonesia. Bagian audio yang relevan adalah dari detik 15.12 hingga 20.87. 🔊: **[Indonesian Speech]** | 🍰: Kemudian untuk voice setting di sini ada lagi di sini kalian bisa menambahkan untuk kecepatannya. |
| **TLoc** | ENG | **Text:** Please give me the start and end time (in seconds) of the following content in the audio clip: *80% equal to 1.38 kilograms.* 🔊: **[English Speech]** | 🍰: The time interval is **[00:22:257 - 00:25:437]**. |
| | | **Text:** Identify when the following content occurs in the audio by providing the start and end time in seconds: *dan sekian informasi dari saya dan kiranya bilamana ada yang ditanyakan silahkan.* 🔊: **[Indonesian Speech]** | 🍰: The start and end time of the content is:\n**Start:** 00:17.060\n**End:** 00:22.259 |
| | SEA | **Text:** : แจ้งช่วงเวลาที่ตรงกับเนื้อหานี้ในไฟล์เสียง (เป็นวินาที): *การออกแบบเสื้อผ้าเครื่องแต่งกาย.* 🔊: **[Thai Speech]** | 🍰: [ 0:0:165 – 0:1:835 ] การออกแบบเสื้อผ้าเครื่องแต่งกาย |
| | | **Text:** Kapan frasa ini muncul di audio? Berikan waktu mulai dan selesai dalam detik: *dan sekian informasi dari saya dan kiranya bilamana ada yang ditanyakan silahkan.* 🔊: **[Indonesian Speech]** | 🍰: Tentu, frasa tersebut muncul pada waktu berikut:\nMulai: 17:039 detik\nSelesai: 22:189 detik |

# F   MODEL-AS-JUDGE PROMPT EXAMPLES

Table 7: Model-as-Judge prompts, adopted from Wang et al. (2024)

| Task | Prompt |
|---|---|
| **SQA** | *[Question]*
*{question}*

*[Reference]*
*{reference}*

*[Model Prediction]*
*{prediction}*

*[Task]*
*Rate the model prediction based on its alignment with the reference, focusing on accuracy and relevance to the reference. Be critical.*
*Score0: The prediction repeats or rephrases the question without giving an answer.*
*Score0: The prediction is refusing to give concrete results, providing something like 'cannot decide'.*
*Score0: The prediction is completely misaligned, providing incorrect or irrelevant information compared to the reference.*
*Score1: The prediction shows minimal alignment, often misunderstanding or providing irrelevant details unrelated to the reference.*
*Score2: The prediction recognizes the topic but diverges significantly from the reference in accuracy or relevance.*
*Score3: The prediction aligns with the reference generally but lacks detail or precise accuracy in some aspects.*
*Score4: The prediction is mostly accurate and relevant, closely following the reference but could be clearer or more detailed.*
*Score5: The prediction is highly accurate, detailed, and matches the reference perfectly, capturing its essence and detail.*

*Your response should be formatted as follows: Explanation: (Provide a concise explanation of your rating, comparing the reference with the model prediction. "The reference is [XXX], while the model prediction is [YYY]. I think ...") Rating: (int)* |
| **AgeR ER GR SpkR** | *[Question]*
*{question}*

*[Reference]*
*{reference}*

*[Model Prediction]*
*{prediction}*

*[Task]*
*Rate the model prediction based on its alignment with the reference, focusing on accuracy and relevance to the reference. Be critical.*
*Score0: The prediction repeats or rephrases the question without giving an answer.*
*Score0: The prediction is refusing to give concrete results, providing something like 'cannot decide'.*
*Score0: The prediction is wrong, providing incorrect or irrelevant information compared to the reference.*
*Score1: The prediction is correct, capturing or covering the meaning from the reference.*

*Your response should be formatted as follows: Explanation: (Provide a concise explanation of your rating, comparing the reference with the model prediction. "The reference is [XXX], while the model prediction is [YYY]. I think ...") Rating: (int)* |

# G    DETAILED EVALUATION RESULTS

Table 8: ASR performance: results across datasets comparing English and SEA prompts. Scores are reported as raw WER and CER values without normalization; lower values indicate better performance.

| Data | Metrics | Lang | Prompt | MERaLiON2 10B | MERaLiON2 3B | SeaLLMs Audio 7B | Phi-4 multi-modal instruct | Kimi Audio | Voxtral mini | Qwen2 Audio 7B Instruct | Qwen 2.5 Omni 3B | Qwen 2.5 Omni 7B | gemma 3n E4B-it | gemma 3n E2B-it | Gemini 2.5 Flash | Whisper large v3 | GPT 4o Audio |
|---|---|---|---|---|---|---|---|---|---|---|---|---|---|---|---|---|---|
| | size | | | 10B | 3B | 7B | 5.6B | 7B | 3B | 7B | 3B | 7B | 4B | 2B | - | 1.5B | - |
| | | | | | | | | | | | | | | | | No Prompt | |
| Bloom-Speech | WER | en | ENG | 0.06 | 0.07 | 0.74 | 0.06 | 0.06 | 0.20 | 0.13 | 0.06 | 0.06 | 0.38 | 0.52 | 0.05 | 0.04 | 0.06 |
| | | | SEA | 0.06 | 0.07 | 0.74 | 0.06 | 0.06 | 0.20 | 0.13 | 0.06 | 0.06 | 0.38 | 0.52 | 0.05 | - | - |
| | CER | my | ENG | 0.74 | 0.99 | 1.08 | 2.29 | 2.50 | 2.94 | 0.98 | 2.74 | 4.19 | 2.36 | 4.94 | 0.73 | 1.55 | 0.98 |
| | | | SEA | 0.78 | 0.98 | 1.19 | 4.82 | 9.57 | 3.33 | 1.72 | 1.30 | 3.24 | 6.56 | 8.36 | 0.75 | - | - |
| | WER | tl | ENG | 0.14 | 0.14 | 0.84 | 5.60 | 9.00 | 3.35 | | 0.41 | 0.50 | 0.23 | 0.63 | 0.09 | 0.12 | 0.11 |
| | | | SEA | 0.17 | 0.17 | 1.37 | 3.10 | 4.06 | 2.24 | 3.10 | 0.41 | 1.84 | 0.58 | 1.48 | 0.08 | - | - |
| Bud500 | WER | vi | ENG | 0.10 | 0.05 | 0.11 | 1.50 | 38.71 | 1.62 | 1.45 | 0.08 | 0.07 | 7.09 | 8.89 | 0.15 | 0.47 | 0.21 |
| | | | SEA | 0.11 | 0.34 | 0.11 | 3.55 | 28.94 | 3.48 | 2.89 | 0.08 | 0.19 | 6.68 | 7.18 | 0.15 | - | - |
| Commonvoice | WER | en | ENG | 0.08 | 0.09 | 0.13 | 0.08 | 0.07 | 0.10 | 0.13 | 0.08 | 0.07 | 0.23 | 0.38 | 0.11 | 0.09 | 0.10 |
| | | | SEA | 0.08 | 0.09 | 0.13 | 0.08 | 0.07 | 0.10 | 0.13 | 0.08 | 0.07 | 0.23 | 0.38 | 0.11 | - | - |
| | WER | id | ENG | 0.11 | 0.11 | 0.10 | 1.44 | 0.68 | 0.38 | 0.74 | 0.11 | 0.10 | 0.47 | 0.25 | 0.03 | 0.08 | 0.06 |
| | | | SEA | 0.07 | 0.13 | 0.19 | 1.61 | 0.60 | 0.37 | 0.79 | 0.12 | 0.10 | 0.68 | 1.40 | 0.03 | - | - |
| | WER | ta | ENG | 0.50 | 0.52 | 1.49 | 1.83 | 4.34 | 1.33 | 1.41 | 1.13 | 1.41 | 0.68 | 1.95 | 0.29 | 0.64 | 0.46 |
| | | | SEA | 0.33 | 0.58 | 1.41 | 3.36 | 2.34 | 1.37 | 1.97 | 1.69 | 1.13 | 2.33 | 2.93 | 0.28 | - | - |
| | CER | th | ENG | 0.16 | 0.1 | 0.05 | 2.35 | 1.38 | 0.53 | 1.31 | 0.14 | 0.76 | 0.51 | 4.30 | 0.03 | 0.07 | 0.07 |
| | | | SEA | 0.09 | 0.21 | 0.05 | 8.41 | 1.14 | 0.45 | 1.27 | 0.16 | 0.78 | 11.41 | 5.79 | 0.04 | - | - |
| | WER | vi | ENG | 0.46 | 0.56 | 0.48 | 1.25 | 0.86 | 0.78 | 1.34 | 0.49 | 0.48 | 1.10 | 1.07 | 0.09 | 0.45 | 0.14 |
| | | | SEA | 0.16 | 0.73 | 0.48 | 1.72 | 1.02 | 0.76 | 0.11 | 0.48 | 0.47 | 1.28 | 0.97 | 0.09 | - | - |
| | CER | zh | ENG | 0.12 | 0.14 | 0.10 | 0.13 | 0.07 | 0.47 | 0.24 | 0.06 | 0.05 | 0.80 | 2.62 | 0.24 | 0.18 | 0.11 |
| | | | SEA | 0.12 | 0.16 | 0.08 | 0.07 | 0.05 | 0.44 | 0.12 | 0.06 | 0.05 | 2.14 | 2.36 | 0.11 | - | - |
| ESD | WER | en | ENG | 0.05 | 0.05 | 0.06 | 0.03 | 0.05 | 0.16 | 0.08 | 0.04 | 0.04 | 0.14 | 0.27 | 0.05 | 0.04 | 0.06 |
| | | | SEA | 0.05 | 0.05 | 0.06 | 0.03 | 0.05 | 0.16 | 0.08 | 0.04 | 0.04 | 0.14 | 0.27 | 0.05 | - | - |
| | CER | zh | ENG | 0.05 | 0.05 | 0.11 | 0.05 | 0.02 | 0.33 | 0.21 | 0.02 | 0.02 | 0.51 | 1.76 | 0.12 | 0.04 | 0.05 |
| | | | SEA | 0.05 | 0.05 | 0.07 | 0.02 | 0.02 | 0.39 | 0.05 | 0.02 | 0.02 | 0.74 | 1.11 | 0.08 | - | - |
| FLEURS | WER | tl | ENG | 0.16 | 0.18 | 1.06 | 5.92 | 0.94 | 1.41 | 1.98 | 0.80 | 0.57 | 0.07 | 0.16 | 0.07 | 0.12 | 0.07 |
| | | | SEA | 0.16 | 0.22 | 1.15 | 4.34 | 0.60 | 1.01 | 1.65 | 0.89 | 1.83 | 0.17 | 0.18 | 0.08 | - | - |
| | WER | id | ENG | 0.06 | 0.10 | 0.45 | 4.07 | 0.47 | 0.34 | 0.74 | 0.18 | 0.10 | 0.54 | 0.09 | 0.03 | 0.07 | 0.04 |
| | | | SEA | 0.06 | 0.12 | 0.12 | 3.19 | 0.36 | 0.28 | 0.64 | 0.11 | 0.31 | 0.10 | 0.09 | 0.03 | - | - |
| | CER | km | ENG | 0.86 | 1.71 | 1.00 | 4.42 | 8.21 | 1.14 | 1.06 | 3.23 | 4.55 | 1.12 | 4.90 | 0.17 | 1.35 | 0.31 |
| | | | SEA | 0.98 | 1.43 | 2.16 | 2.25 | 4.42 | 0.92 | 1.40 | 11.63 | 4.48 | 2.41 | 7.83 | 0.20 | - | - |
| | CER | lo | ENG | 0.39 | 0.86 | 1.01 | 2.45 | 4.52 | 1.29 | 1.08 | 1.39 | 2.63 | 0.15 | 1.42 | 0.16 | 1.02 | 0.38 |
| | | | SEA | 0.51 | 0.81 | 1.36 | 1.53 | 6.34 | 1.24 | 1.20 | 1.44 | 1.80 | 1.75 | 2.60 | 0.22 | - | - |
| | WER | ms | ENG | 0.11 | 0.14 | 0.76 | 3.34 | 1.32 | 0.44 | 1.03 | 0.28 | 0.27 | 0.86 | 0.18 | 0.05 | 0.08 | 0.06 |
| | | | SEA | 0.11 | 0.19 | 0.29 | 2.74 | 0.35 | 0.42 | 0.87 | 0.30 | 0.17 | 0.15 | 0.15 | 0.05 | - | - |
| | CER | my | ENG | 0.79 | 1.19 | 1.10 | 1.85 | 10.20 | 3.62 | 1.07 | 2.62 | 4.43 | 0.15 | 3.16 | 0.13 | 1.16 | 0.44 |
| | | | SEA | 0.76 | 1.03 | 1.10 | 4.35 | 8.87 | 3.07 | 1.11 | 1.99 | 4.01 | 1.42 | 1.97 | 0.12 | - | - |
| | CER | th | ENG | 0.14 | 0.13 | 0.10 | 4.14 | 2.93 | 0.57 | 1.04 | 0.16 | 1.20 | 0.15 | 4.03 | 0.07 | 0.09 | 0.04 |
| | | | SEA | 0.11 | 0.22 | 0.10 | 5.40 | 2.45 | 0.49 | 1.14 | 0.15 | 0.95 | 1.42 | 4.73 | 0.04 | - | - |
| | WER | vi | ENG | 0.10 | 0.16 | 0.91 | 4.46 | 2.39 | 0.40 | 1.05 | 0.10 | 0.09 | 0.17 | 0.30 | 0.06 | 0.08 | 0.04 |
| | | | SEA | 0.08 | 0.26 | 0.16 | 4.11 | 1.53 | 0.34 | 1.42 | 0.11 | 0.09 | 0.19 | 0.32 | 0.04 | - | - |
| | CER | zh | ENG | 0.10 | 0.12 | 0.84 | 0.1 | 0.08 | 0.36 | 0.18 | 0.07 | 0.06 | 0.25 | 1.00 | 0.18 | 0.08 | 0.07 |
| | | | SEA | 0.10 | 0.11 | 0.84 | 0.07 | 0.08 | 0.35 | 0.10 | 0.07 | 0.06 | 1.00 | 1.00 | 0.09 | - | - |
| ASR-MALCSC | WER | ms | ENG | 0.21 | 0.26 | 0.79 | 3.82 | 14.23 | 1.30 | 1.77 | 0.21 | 1.01 | 4.86 | 2.81 | 0.26 | 0.31 | 0.42 |
| | | | SEA | 0.22 | 0.27 | 0.71 | 2.66 | 7.70 | 1.84 | 1.68 | 0.71 | 1.24 | 4.07 | 4.58 | 0.26 | - | - |
| MIG | CER | my | ENG | 1.06 | 1.59 | 1.47 | 2.58 | 12.47 | 8.30 | 1.47 | 1.12 | 9.45 | 8.13 | 1.63 | 0.31 | 2.55 | 0.78 |
| | | | SEA | 0.97 | 1.53 | 2.09 | 9.35 | 15.58 | 2.80 | 2.09 | 1.64 | 4.43 | 3.58 | 5.53 | 0.29 | - | - |
| OpenSLR | CER | km | ENG | 0.68 | 1.72 | 1.14 | 1.80 | 2.33 | 2.29 | 1.10 | 2.71 | 1.61 | 2.29 | 2.94 | 0.09 | 4.70 | 0.29 |
| | | | SEA | 0.88 | 1.74 | 2.18 | 5.91 | 4.64 | 2.80 | 1.13 | 8.42 | 1.70 | 2.89 | 3.92 | 0.30 | - | - |
| | CER | my | ENG | 0.71 | 0.97 | 1.16 | 1.80 | 6.78 | 5.34 | 1.16 | 1.57 | 3.88 | 1.26 | 1.51 | 0.05 | 2.51 | 0.42 |
| | | | SEA | 0.74 | 1.07 | 2.63 | 7.51 | 13.1 | 5.52 | 2.13 | 0.97 | 4.40 | 0.99 | 1.42 | 0.06 | - | - |
| | WER | ta | ENG | 0.26 | 0.34 | 1.55 | 2.07 | 4.58 | 1.06 | 1.55 | 1.58 | 1.26 | 0.22 | 0.45 | 0.24 | 2.51 | 0.35 |
| | | | SEA | 0.26 | 0.40 | 1.30 | 3.59 | 2.47 | 1.10 | 2.22 | 1.58 | 1.47 | 0.71 | 0.85 | 0.25 | - | - |
| SFDUSC | WER | tl | ENG | 0.24 | 0.27 | 1.20 | 3.77 | 0.31 | 2.44 | 2.48 | 0.59 | 0.59 | 0.42 | 0.48 | 0.23 | 0.25 | 0.24 |
| | | | SEA | 0.25 | 0.10 | 1.11 | 3.82 | 1.30 | 2.24 | 4.07 | 0.70 | 0.61 | 0.39 | 0.75 | 0.21 | - | - |
| ASR-SgpCCSC | CER | zh | ENG | 0.07 | 0.11 | 0.30 | 0.13 | 0.04 | 0.62 | 0.19 | 0.05 | 0.05 | 1.19 | 0.80 | 0.19 | 1.00 | 0.13 |
| | | | SEA | 0.07 | 0.09 | 0.21 | 3.82 | 0.04 | 0.71 | 0.21 | 0.05 | 0.04 | 5.55 | 1.77 | 0.12 | - | - |
| SG Streets | WER | en | ENG | 0.10 | 0.05 | 0.57 | 0.17 | 0.82 | 0.39 | 0.20 | 0.09 | 0.09 | 2.27 | 4.11 | 0.22 | 1.00 | 0.29 |
| | | | SEA | 0.10 | 0.05 | 0.57 | 0.17 | 0.82 | 0.39 | 0.20 | 0.09 | 0.09 | 2.27 | 4.11 | 0.22 | - | - |
| ASR-SMalDuSC | WER | ms | ENG | 0.07 | 0.11 | 0.27 | 2.28 | 2.21 | 0.55 | 0.27 | 0.21 | 0.22 | 0.23 | 0.18 | 0.02 | 1.00 | 0.03 |
| | | | SEA | 0.07 | 0.10 | 0.27 | 2.24 | 1.30 | 0.56 | 1.08 | 0.21 | 0.23 | 0.26 | 0.24 | 0.02 | - | - |
| Thai Elderly Speech | CER | th | ENG | 0.08 | 0.07 | 0.04 | 2.07 | 1.56 | 0.71 | 1.35 | 0.14 | 0.17 | 0.54 | 2.72 | 0.06 | 0.06 | 0.10 |
| | | | SEA | 0.08 | 0.22 | 0.04 | 8.38 | 1.16 | 0.75 | 1.32 | 0.20 | 0.12 | 7.46 | 2.63 | 0.05 | - | - |
| LOTUS | CER | th | ENG | 0.02 | 0.03 | 0.02 | 3.35 | 4.51 | 0.46 | 1.14 | 0.04 | 0.03 | 0.06 | 1.06 | 0.01 | 0.03 | 0.01 |
| | | | SEA | 0.02 | 0.12 | 0.03 | 6.51 | 1.37 | 0.50 | 0.03 | 0.04 | 0.03 | 1.25 | 1.17 | 0.03 | - | - |
| VietMed | WER | vi | ENG | 0.27 | 0.64 | 0.27 | 4.47 | 21.47 | 0.92 | 0.27 | 0.40 | 0.35 | 2.49 | 4.80 | 0.18 | 0.65 | 0.44 |
| | | | SEA | 0.28 | 0.49 | 0.27 | 3.23 | 19.44 | 1.48 | 1.56 | 0.40 | 0.26 | 2.71 | 4.07 | 0.19 | - | - |
| Average | | | ENG | 0.27 | 0.41 | 0.65 | 2.29 | 4.85 | 1.42 | 0.95 | 0.67 | 1.24 | 1.27 | 2.01 | 0.14 | 0.74 | 0.22 |
| | | | SEA | 0.27 | 0.43 | 0.74 | 3.39 | 4.30 | 1.45 | 1.20 | 1.11 | 1.08 | 2.28 | 2.50 | 0.14 | - | - |

Table 9: Age Recognition performance: results across datasets comparing English and SEA prompts. Evaluation is conducted using the Model-as-Judge metric, where higher scores indicate better performance.

| Model | | | MERaLiON2 10B | MERaLiON2 3B | SeaLLMs Audio 7B | Phi-4 multi-modal instruct | Kimi Audio | Voxtral mini | Qwen2 Audio 7B Instruct | Qwen 2.5 Omni 3B | Qwen 2.5 Omni 7B | gemma 3n E4B-it | gemma 3n E2B-it | Gemini 2.5 Flash | GPT 4o Audio |
|---|---|---|---|---|---|---|---|---|---|---|---|---|---|---|---|
| Size | | | 10B | 3B | 7B | 5.6B | 7B | 3B | 7B | 3B | 7B | 4B | 2B | - | - |
| **Data** | **Lang** | **Prompt** | | | | | | | | | | | | | |
| **Commonvoice** | en | ENG | 63.10 | 61.00 | 54.10 | 45.35 | 38.00 | 58.10 | 36.90 | 29.30 | 36.90 | 63.60 | 59.45 | 72.00 | 65.00 |
| | | SEA | 63.10 | 61.00 | 54.10 | 45.35 | 38.00 | 58.10 | 36.90 | 29.30 | 36.90 | 63.60 | 59.45 | 72.00 | 65.00 |
| | ta | ENG | 64.65 | 71.60 | 52.70 | 49.20 | 51.70 | 77.00 | 18.00 | 48.50 | 18.00 | 65.50 | 63.95 | 75.00 | 71.00 |
| | | SEA | 47.90 | 16.10 | 1.10 | 16.30 | 14.80 | 38.10 | 19.35 | 7.00 | 19.35 | 38.40 | 17.00 | 77.00 | 62.00 |
| | th | ENG | 57.81 | 67.23 | 68.77 | 40.39 | 49.10 | 81.81 | 16.90 | 48.52 | 16.90 | 73.42 | 72.13 | 74.00 | 78.00 |
| | | SEA | 42.19 | 51.87 | 1.00 | 53.16 | 63.48 | 61.42 | 0.25 | 43.74 | 0.26 | 68.13 | 49.16 | 85.00 | 53.00 |
| | vi | ENG | 73.23 | 66.57 | 63.03 | 34.81 | 55.52 | 85.83 | 6.96 | 58.58 | 6.96 | 71.79 | 66.03 | 86.00 | 86.00 |
| | | SEA | 64.35 | 52.34 | 1.00 | 30.09 | 62.61 | 33.73 | 1.56 | 33.13 | 1.56 | 61.40 | 33.01 | 90.00 | 81.00 |
| | zh | ENG | 72.40 | 74.60 | 77.20 | 47.50 | 45.25 | 75.80 | 22.30 | 67.60 | 22.00 | 74.10 | 68.00 | 73.00 | 83.00 |
| | | SEA | 69.00 | 51.00 | 1.00 | 65.00 | 63.20 | 71.70 | 57.60 | 59.20 | 57.60 | 62.15 | 35.20 | 73.00 | 45.00 |
| **Average** | | ENG | 66.24 | 68.20 | 63.16 | 43.45 | 47.91 | 75.71 | 20.21 | 50.50 | 20.15 | 69.68 | 65.91 | 76.00 | 76.60 |
| | | SEA | 57.31 | 46.46 | 11.64 | 41.98 | 48.42 | 52.61 | 23.13 | 34.47 | 23.13 | 58.74 | 38.76 | 79.40 | 61.20 |

Table 10: Emotion Recognition performance: results across datasets comparing English and SEA prompts. Evaluation is conducted using the Model-as-Judge metric, where higher scores indicate better performance.

| Model | | | MERaLiON2 10B | MERaLiON2 3B | SeaLLMs Audio 7B | Phi-4 multi-modal instruct | Kimi Audio | Voxtral mini | Qwen2 Audio 7B Instruct | Qwen 2.5 Omni 3B | Qwen 2.5 Omni 7B | gemma 3n E4B-it | gemma 3n E2B-it | Gemini 2.5 Flash | GPT 4o Audio |
|---|---|---|---|---|---|---|---|---|---|---|---|---|---|---|---|
| Size | | | 10B | 3B | 7B | 5.6B | 7B | 3B | 7B | 3B | 7B | 4B | 2B | - | - |
| **Data** | | **Lang Prompt** | | | | | | | | | | | | | |
| **EmoTa** | ta | ENG | 8.33 | 15.12 | 6.78 | 21.58 | 15.33 | 8.33 | 23.08 | 7.26 | 14.58 | 8.49 | 8.23 | 12.00 | 8.00 |
| | | SEA | 11.97 | 10.84 | 0.85 | 2.24 | 7.91 | 3.73 | 0.96 | 0.85 | 1.00 | 6.94 | 11.43 | 9.00 | 17.00 |
| **ESD** | en | ENG | 19.50 | 23.40 | 12.85 | 19.80 | 64.70 | 9.05 | 40.90 | 9.40 | 13.55 | 8.50 | 9.15 | 15.00 | 13.00 |
| | | SEA | 19.50 | 23.40 | 12.85 | 19.80 | 64.70 | 9.05 | 40.90 | 9.40 | 13.55 | 8.50 | 9.15 | 15.00 | 13.00 |
| | zh | ENG | 16.70 | 19.90 | 7.70 | 17.80 | 71.25 | 5.55 | 34.60 | 15.50 | 18.10 | 10.55 | 11.60 | 14.00 | 7.00 |
| | | SEA | 14.15 | 18.75 | 7.45 | 12.50 | 66.15 | 4.80 | 47.55 | 17.70 | 16.10 | 12.65 | 13.65 | 14.00 | 13.50 |
| **IndoWaveSentiment** | id | ENG | 20.00 | 20.67 | 11.67 | 18.33 | 27.83 | 11.00 | 13.33 | 11.33 | 13.33 | 17.00 | 15.83 | 26.00 | 23.00 |
| | | SEA | 23.33 | 18.00 | 17.50 | 3.67 | 29.83 | 8.67 | 15.50 | 16.00 | 14.33 | 13.33 | 9.67 | 18.00 | 22.00 |
| **M3ED** | zh | ENG | 19.35 | 26.65 | 10.90 | 20.65 | 19.00 | 7.95 | 14.30 | 11.95 | 13.50 | 7.35 | 10.20 | 16.00 | 12.00 |
| | | SEA | 23.25 | 23.40 | 13.35 | 13.50 | 23.15 | 0.85 | 19.80 | 18.55 | 19.60 | 13.20 | 10.75 | 16.50 | 14.00 |
| **TEC** | ta | ENG | 34.85 | 42.42 | 24.85 | 30.30 | 42.42 | 23.12 | 30.91 | 28.48 | 27.58 | 21.52 | 19.39 | 42.50 | 43.00 |
| | | SEA | 36.97 | 20.30 | 1.82 | 3.64 | 21.81 | 6.06 | 2.42 | 1.21 | 1.21 | 31.52 | 27.88 | 35.00 | 47.00 |
| **THAI SER** | th | ENG | 12.36 | 19.74 | 11.62 | 17.59 | 17.48 | 9.31 | 14.19 | 10.26 | 13.66 | 13.82 | 11.10 | 11.00 | 13.00 |
| | | SEA | 13.19 | 16.44 | 10.37 | 9.06 | 84.29 | 4.29 | 8.38 | 5.97 | 7.33 | 11.36 | 9.63 | 10.00 | 10.00 |
| **Average** | | ENG | 22.56 | 26.55 | 15.53 | 22.99 | 35.88 | 14.55 | 26.74 | 15.79 | 19.10 | 17.61 | 16.75 | 23.81 | 19.88 |
| | | SEA | 23.97 | 21.96 | 12.76 | 12.78 | 40.86 | 9.94 | 22.26 | 12.72 | 13.95 | 18.90 | 17.59 | 21.44 | 22.06 |

Table 11: Gender Recognition performance: results across datasets comparing English and SEA prompts. Evaluation is conducted using the Model-as-Judge metric, where higher scores indicate better performance.

| Model | | | MERaLiON2 10B | MERaLiON2 3B | SeaLLMs Audio 7B | Phi-4 multi-modal instruct | Kimi Audio | Voxtral mini | Qwen2 Audio 7B Instruct | Qwen 2.5 Omni 3B | Qwen 2.5 Omni 7B | gemma 3n E4B-it | gemma 3n E2B-it | Gemini 2.5 Flash | GPT 4o Audio |
|---|---|---|---|---|---|---|---|---|---|---|---|---|---|---|---|
| Size | | | 10B | 3B | 7B | 5.6B | 7B | 3B | 7B | 3B | 7B | 4B | 2B | - | - |
| **Data** | **Lang** | **Prompt** | | | | | | | | | | | | | |
| **Commonvoice** | id | ENG | 45.20 | 45.30 | 54.90 | 35.80 | 93.50 | 32.00 | 95.20 | 41.20 | 59.30 | 49.60 | 22.20 | 95.00 | 46.00 |
| | | SEA | 57.30 | 62.92 | 56.18 | 57.30 | 66.30 | 44.94 | 61.80 | 62.92 | 52.81 | 57.30 | 60.67 | 82.02 | 53.93 |
| | ta | ENG | 53.00 | 51.40 | 50.10 | 45.70 | 90.60 | 44.60 | 96.80 | 23.80 | 53.90 | 54.70 | 37.20 | 95.00 | 33.00 |
| | | SEA | 40.40 | 46.90 | 0.10 | 7.90 | 57.60 | 13.20 | 7.40 | 0.40 | 4.10 | 12.30 | 17.20 | 92.00 | 35.00 |
| | th | ENG | 50.07 | 36.14 | 57.43 | 28.51 | 96.12 | 44.18 | 96.79 | 46.18 | 61.45 | 51.54 | 29.18 | 93.00 | 50.00 |
| | | SEA | 23.96 | 53.55 | 49.00 | 28.92 | 89.42 | 3.88 | 96.39 | 32.33 | 67.47 | 19.14 | 2.54 | 91.00 | 40.00 |
| | vi | ENG | 24.05 | 18.82 | 83.01 | 34.25 | 94.51 | 50.07 | 95.16 | 42.75 | 49.41 | 38.04 | 33.73 | 90.00 | 26.00 |
| | | SEA | 14.64 | 43.14 | 81.57 | 18.30 | 94.41 | 22.09 | 97.52 | 21.44 | 9.80 | 3.66 | 3.53 | 77.00 | 35.00 |
| | zh | ENG | 53.70 | 34.60 | 68.90 | 57.10 | 83.90 | 40.90 | 98.20 | 75.30 | 81.80 | 41.00 | 17.80 | 90.00 | 49.00 |
| | | SEA | 35.50 | 36.50 | 68.90 | 44.70 | 64.50 | 31.40 | 98.40 | 60.90 | 88.90 | 37.95 | 26.60 | 91.00 | 21.00 |
| **EmoTa** | ta | ENG | 67.31 | 49.36 | 52.88 | 40.81 | 90.92 | 22.01 | 98.82 | 17.41 | 36.81 | 55.34 | 49.36 | 94.00 | 25.00 |
| | | SEA | 48.93 | 47.76 | 0.21 | 8.65 | 75.43 | 10.47 | 2.35 | 1.28 | 2.67 | 11.65 | 9.19 | 94.00 | 33.00 |
| **FLEURS** | en | ENG | 58.27 | 67.54 | 39.41 | 62.44 | 97.84 | 8.04 | 99.38 | 44.20 | 30.76 | 52.24 | 26.35 | 97.00 | 78.00 |
| | | SEA | 58.27 | 67.54 | 39.41 | 62.44 | 97.84 | 8.04 | 99.38 | 44.20 | 30.76 | 52.24 | 26.35 | 97.00 | 78.00 |
| | km | ENG | 56.60 | 31.90 | 72.03 | 29.93 | 99.22 | 38.04 | 78.82 | 33.73 | 43.66 | 48.89 | 11.37 | 99.00 | 62.00 |
| | | SEA | 43.40 | 56.34 | 0.39 | 26.08 | 47.06 | 1.96 | 5.49 | 2.35 | 1.05 | 2.75 | 0.78 | 97.00 | 15.00 |
| **IndoWaveSentiment** | id | ENG | 71.67 | 63.67 | 50.33 | 42.00 | 91.33 | 5.33 | 98.33 | 13.00 | 65.00 | 58.00 | 20.00 | 96.00 | 60.00 |
| | | SEA | 60.67 | 3.60 | 50.00 | 22.33 | 79.33 | 1.33 | 94.00 | 19.33 | 31.67 | 24.00 | 13.33 | 95.00 | 14.00 |
| **M3ED** | zh | ENG | 84.30 | 64.50 | 51.40 | 82.80 | 74.60 | 8.80 | 86.30 | 15.80 | 43.30 | 58.05 | 30.80 | 84.00 | 23.00 |
| | | SEA | 70.70 | 80.60 | 53.80 | 72.10 | 78.90 | 6.10 | 78.00 | 11.30 | 78.90 | 45.45 | 24.20 | 81.00 | 12.00 |
| **OpenSLR** | ta | ENG | 55.30 | 50.50 | 50.30 | 47.00 | 97.50 | 34.00 | 99.10 | 41.10 | 40.50 | 52.00 | 29.10 | 97.00 | 47.00 |
| | | SEA | 37.80 | 42.20 | 0.10 | 6.20 | 60.80 | 9.40 | 8.00 | 0.80 | 4.70 | 15.10 | 14.60 | 97.00 | 36.00 |
| **SG Streets** | en | ENG | 89.63 | 39.43 | 61.99 | 53.46 | 95.94 | 5.49 | 84.55 | 8.94 | 5.69 | 64.23 | 31.30 | 98.00 | 32.00 |
| | | SEA | 89.63 | 39.43 | 61.99 | 53.46 | 95.94 | 5.49 | 84.55 | 8.94 | 5.69 | 64.23 | 31.30 | 98.00 | 32.00 |
| **ASR-SMalDuSC** | ms | ENG | 52.40 | 49.80 | 52.20 | 22.30 | 96.70 | 26.80 | 93.90 | 66.50 | 58.90 | 49.70 | 28.00 | 98.00 | 76.00 |
| | | SEA | 44.00 | 50.30 | 45.70 | 37.00 | 94.30 | 19.00 | 61.30 | 23.55 | 15.90 | 34.70 | 20.20 | 96.00 | 24.00 |
| **Thai Elderly Speech** | th | ENG | 68.15 | 55.75 | 34.48 | 22.58 | 94.96 | 44.76 | 97.48 | 45.16 | 60.28 | 65.73 | 26.66 | 99.00 | 46.00 |
| | | SEA | 26.92 | 46.47 | 69.56 | 45.67 | 84.68 | 3.33 | 96.98 | 23.99 | 82.01 | 37.40 | 2.42 | 85.00 | 51.00 |
| **THAI SER** | th | ENG | 63.46 | 63.87 | 49.21 | 53.40 | 86.81 | 18.32 | 87.54 | 31.10 | 60.00 | 56.86 | 27.23 | 86.00 | 44.00 |
| | | SEA | 61.78 | 58.12 | 81.15 | 41.68 | 84.29 | 12.14 | 85.86 | 26.13 | 80.00 | 33.93 | 3.77 | 85.00 | 34.00 |
| **Vietnam-Celeb** | vi | ENG | 65.80 | 54.80 | 50.90 | 49.30 | 71.10 | 19.10 | 69.70 | 31.70 | 56.60 | 57.80 | 23.10 | 69.00 | 41.00 |
| | | SEA | 61.40 | 63.50 | 50.30 | 30.30 | 72.80 | 5.00 | 69.30 | 23.00 | 27.50 | 10.65 | 1.60 | 73.00 | 36.00 |
| **Average** | | ENG | 59.78 | 49.43 | 55.04 | 44.98 | 89.52 | 28.67 | 90.46 | 37.69 | 50.60 | 53.59 | 29.65 | 91.88 | 46.58 |
| | | SEA | 46.93 | 47.33 | 44.40 | 34.05 | 78.47 | 11.95 | 66.05 | 22.70 | 35.65 | 29.00 | 17.36 | 89.59 | 32.94 |

Table 12: Speaker Recognition performance: results across datasets comparing English and SEA prompts. Evaluation is conducted using the Model-as-Judge metric, where higher scores indicate better performance.

| Model | | | MERaLiON2 10B | MERaLiON2 3B | SeaLLMs Audio 7B | Phi-4 multi-modal instruct | Kimi Audio | Voxtral mini | Qwen2 Audio 7B Instruct | Qwen 2.5 Omni 3B | Qwen 2.5 Omni 7B | gemma 3n E4B-it | gemma 3n E2B-it | Gemini 2.5 Flash | GPT 4o Audio |
|---|---|---|---|---|---|---|---|---|---|---|---|---|---|---|---|
| Size | | | 10B | 3B | 7B | 5.6B | 7B | 3B | 7B | 3B | 7B | 4B | 2B | - | - |
| **Data** | **Lang** | **Prompt** | | | | | | | | | | | | | |
| **EmoTa** | ta | ENG | 53.10 | 43.06 | 56.94 | 14.10 | 57.05 | 46.15 | 50.64 | 26.50 | 13.25 | 39.53 | 41.99 | 71.00 | 6.00 |
| | | SEA | 53.85 | 31.73 | 0.00 | 15.71 | 51.17 | 33.97 | 15.44 | 7.91 | 12.39 | 27.99 | 40.92 | 73.00 | 13.00 |
| **ESD** | en | ENG | 52.90 | 39.20 | 56.75 | 34.90 | 53.80 | 37.70 | 47.90 | 24.40 | 5.70 | 39.10 | 45.90 | 71.00 | 10.00 |
| | | SEA | 52.90 | 39.20 | 56.75 | 34.90 | 53.80 | 37.70 | 47.90 | 24.40 | 5.70 | 39.10 | 45.90 | 71.00 | 10.00 |
| | zh | ENG | 53.10 | 43.70 | 49.10 | 24.80 | 48.00 | 42.40 | 51.60 | 31.70 | 23.30 | 41.40 | 41.20 | 59.00 | 12.00 |
| | | SEA | 50.60 | 43.30 | 48.10 | 45.20 | 44.90 | 43.50 | 45.30 | 38.10 | 30.70 | 45.58 | 44.40 | 61.00 | 4.00 |
| **mig** | my | ENG | 50.00 | 35.15 | 53.50 | 22.50 | 50.90 | 31.50 | 54.60 | 30.60 | 13.50 | 35.90 | 29.40 | 70.00 | 5.00 |
| | | SEA | 36.50 | 12.70 | 0.20 | 11.60 | 12.60 | 27.20 | 1.00 | 6.30 | 6.45 | 43.80 | 37.30 | 77.00 | 4.00 |
| **ASR-SMalDuSC** | ms | ENG | 59.20 | 44.50 | 57.80 | 23.20 | 59.30 | 47.00 | 46.60 | 24.00 | 10.80 | 37.40 | 44.00 | 74.00 | 14.00 |
| | | SEA | 48.05 | 39.30 | 42.20 | 28.20 | 59.70 | 44.70 | 41.10 | 29.40 | 17.20 | 47.60 | 38.90 | 82.00 | 16.00 |
| **Thai Elderly Speech** | th | ENG | 52.92 | 38.13 | 52.29 | 20.42 | 52.10 | 50.31 | 48.02 | 29.48 | 14.27 | 42.81 | 43.65 | 75.00 | 6.00 |
| | | SEA | 34.06 | 26.77 | 37.92 | 27.40 | 56.25 | 41.56 | 43.65 | 6.56 | 5.00 | 38.49 | 30.52 | 71.00 | 7.00 |
| **THAI SER** | th | ENG | 51.02 | 38.45 | 50.70 | 22.13 | 58.00 | 46.72 | 50.16 | 30.93 | 18.80 | 41.57 | 46.94 | 70.00 | 20.00 |
| | | SEA | 42.96 | 36.84 | 42.86 | 30.08 | 60.47 | 34.80 | 42.11 | 12.24 | 6.20 | 37.16 | 35.02 | 61.00 | 7.00 |
| **VoxVietnam-O** | vi | ENG | 53.80 | 35.80 | 51.60 | 24.10 | 54.10 | 41.40 | 49.30 | 31.10 | 17.40 | 41.30 | 45.70 | 79.00 | 11.00 |
| | | SEA | 50.00 | 39.80 | 46.90 | 26.80 | 46.40 | 25.30 | 56.00 | 17.40 | 7.90 | 38.00 | 41.60 | 78.00 | 6.00 |
| **Average** | | ENG | 53.25 | 39.75 | 53.59 | 23.27 | 54.16 | 42.90 | 49.85 | 28.59 | 14.63 | 39.88 | 42.35 | 71.13 | 10.50 |
| | | SEA | 46.12 | 33.71 | 34.37 | 27.48 | 48.16 | 36.09 | 36.56 | 17.79 | 11.44 | 39.72 | 39.32 | 71.75 | 8.38 |

Table 13: Speech Translation performance: results across datasets comparing English and SEA prompts. Evaluation is based on the BLEU metric, where higher scores indicate better performance.

| Model | | | MERaLiON2 10B | MERaLiON2 3B | SeaLLMs Audio 7B | Phi-4 multi-modal instruct | Kimi Audio | Voxtral mini | Qwen2 Audio 7B Instruct | Qwen 2.5 Omni 3B | Qwen 2.5 Omni 7B | gemma 3n E4B-it | gemma 3n E2B-it | Gemini 2.5 Flash | GPT 4o Audio |
|---|---|---|---|---|---|---|---|---|---|---|---|---|---|---|---|
| Size | | | 10B | 3B | 7B | 5.6B | 7B | 3B | 7B | 3B | 7B | 4B | 2B | - | - |
| **Data** | **Lang** | **Prompt** | | | | | | | | | | | | | |
| **FLEURS** | id | ENG | 32.48 | 20.97 | 25.55 | 0.70 | 10.75 | 35.02 | 9.37 | 16.65 | 16.55 | 21.40 | 20.52 | 24.07 | 33.44 |
| | | SEA | 30.02 | 21.98 | 25.97 | 0.13 | 16.81 | 34.58 | 6.23 | 11.15 | 17.93 | 24.80 | 17.57 | 24.43 | 34.90 |
| | km | ENG | 2.19 | 0.50 | 0.73 | 0.23 | 0.35 | 7.04 | 0.47 | 0.26 | 0.48 | 4.72 | 2.52 | 12.40 | 6.34 |
| | | SEA | 1.97 | 0.45 | 0.33 | 0.05 | 0.49 | 0.08 | 0.11 | 0.27 | 0.24 | 4.71 | 2.50 | 13.76 | 7.52 |
| | lo | ENG | 11.34 | 1.03 | 5.91 | 0.04 | 0.55 | 10.44 | 0.47 | 2.49 | 3.06 | 7.66 | 6.75 | 15.36 | 13.61 |
| | | SEA | 12.09 | 0.88 | 1.04 | 0.01 | 1.04 | 0.55 | 0.11 | 0.80 | 2.35 | 8.94 | 7.18 | 15.04 | 13.06 |
| | ms | ENG | 31.04 | 15.52 | 19.71 | 1.04 | 6.15 | 30.44 | 5.25 | 12.16 | 13.19 | 18.24 | 16.41 | 26.49 | 33.53 |
| | | SEA | 34.35 | 13.09 | 20.30 | 0.22 | 20.02 | 30.33 | 2.57 | 10.12 | 14.24 | 23.04 | 14.37 | 23.70 | 33.93 |
| | my | ENG | 0.37 | 0.12 | 0.29 | 0.11 | 0.03 | 0.73 | 0.40 | 0.11 | 0.35 | 0.78 | 0.35 | 8.59 | 1.47 |
| | | SEA | 0.59 | 0.14 | 0.03 | 0.11 | 0.04 | 0.63 | 0.14 | 0.02 | 0.10 | 0.73 | 0.10 | 15.05 | 2.08 |
| | th | ENG | 17.22 | 3.47 | 12.30 | 0.14 | 2.42 | 19.53 | 0.47 | 7.90 | 8.34 | 12.95 | 9.57 | 16.93 | 22.41 |
| | | SEA | 18.70 | 5.59 | 11.51 | 0.02 | 3.52 | 20.15 | 0.55 | 7.86 | 9.29 | 14.90 | 8.49 | 20.25 | 22.93 |
| | tl | ENG | 25.79 | 8.28 | 1.67 | 0.92 | 5.99 | 30.42 | 1.68 | 2.52 | 2.50 | 17.14 | 12.98 | 14.50 | 28.48 |
| | | SEA | 27.22 | 7.76 | 1.76 | 0.10 | 23.27 | 30.69 | 1.25 | 0.89 | 2.29 | 22.47 | 14.44 | 16.09 | 26.20 |
| | vi | ENG | 19.18 | 6.50 | 13.52 | 0.12 | 3.94 | 24.79 | 2.05 | 11.40 | 12.02 | 7.19 | 4.89 | 17.26 | 27.10 |
| | | SEA | 22.91 | 5.95 | 13.00 | 0.01 | 4.19 | 25.05 | 1.91 | 10.31 | 12.32 | 11.84 | 7.51 | 19.74 | 30.27 |
| | zh | ENG | 20.14 | 11.63 | 17.01 | 24.08 | 0.01 | 21.36 | 20.69 | 14.89 | 14.73 | 8.72 | 6.75 | 16.18 | 24.77 |
| | | SEA | 20.74 | 10.94 | 17.11 | 2.28 | 0.01 | 21.30 | 15.88 | 15.02 | 13.81 | 10.78 | 6.37 | 21.91 | 21.60 |
| **Average** | | ENG | 17.75 | 7.56 | 10.74 | 3.04 | 3.36 | 19.98 | 4.54 | 7.60 | 7.91 | 10.98 | 8.97 | 16.86 | 21.24 |
| | | SEA | 18.73 | 7.42 | 10.11 | 0.32 | 7.71 | 18.15 | 3.20 | 6.27 | 8.06 | 13.58 | 8.72 | 18.89 | 21.39 |

Table 14: Speech Question Answering performance: results across datasets comparing English and SEA prompts. Evaluation is based on Model-as-Judge matric, where higher scores indicate better performance.

| Model | | | MERaLiON2 10B | MERaLiON2 3B | SeaLLMs Audio 7B | Phi-4 multi-modal instruct | Kimi Audio | Voxtral mini | Qwen2 Audio 7B Instruct | Qwen 2.5 Omni 3B | Qwen 2.5 Omni 7B | gemma 3n E4B-it | gemma 3n E2B-it | Gemini 2.5 Flash | GPT 4o Audio |
|---|---|---|---|---|---|---|---|---|---|---|---|---|---|---|---|
| Size | | | 10B | 3B | 7B | 5.6B | 7B | 3B | 7B | 3B | 7B | 4B | 2B | - | - |
| **Data** | **Lang** | **Prompt** | | | | | | | | | | | | | |
| **ASR-SgpCCSC** | zh | ENG | 78.59 | 65.94 | 73.85 | 76.83 | 61.81 | 73.80 | 73.60 | 85.59 | 66.35 | 70.33 | 61.31 | 84.20 | 77.80 |
| | | SEA | 84.33 | 69.97 | 74.81 | 80.15 | 55.11 | 76.32 | 76.62 | 72.19 | 61.52 | 72.29 | 64.28 | 88.00 | 77.80 |
| **SG Streets** | en | ENG | 86.11 | 82.95 | 82.31 | 81.89 | 82.53 | 87.37 | 80.63 | 70.98 | 61.68 | 73.05 | 72.00 | 91.37 | 82.95 |
| | | SEA | 86.11 | 82.95 | 82.31 | 81.89 | 82.52 | 87.37 | 80.63 | 70.98 | 61.68 | 73.05 | 72.00 | 91.37 | 82.95 |
| **YODAS2** | en | ENG | 84.70 | 76.05 | 77.00 | 81.71 | 75.91 | 84.58 | 74.72 | 71.69 | 70.67 | 85.54 | 80.83 | 85.20 | 80.60 |
| | | SEA | 84.70 | 76.05 | 77.00 | 81.71 | 75.91 | 84.58 | 74.72 | 71.69 | 70.67 | 85.54 | 80.83 | 85.20 | 80.60 |
| | id | ENG | 74.73 | 53.80 | 68.41 | 30.60 | 58.77 | 70.93 | 45.79 | 62.94 | 61.51 | 74.99 | 70.91 | 83.80 | 73.40 |
| | | SEA | 81.45 | 63.94 | 74.65 | 22.66 | 61.83 | 75.96 | 51.11 | 63.64 | 63.98 | 79.20 | 74.27 | 85.40 | 77.40 |
| | th | ENG | 73.13 | 49.82 | 66.53 | 40.58 | 60.00 | 71.34 | 37.74 | 62.85 | 62.69 | 77.29 | 71.92 | 89.00 | 75.00 |
| | | SEA | 78.30 | 45.39 | 75.09 | 16.43 | 55.07 | 71.78 | 30.12 | 60.18 | 71.10 | 80.80 | 75.97 | 94.00 | 76.60 |
| | vi | ENG | 62.56 | 38.75 | 58.19 | 30.34 | 50.67 | 65.67 | 34.54 | 53.15 | 52.08 | 61.96 | 55.58 | 77.00 | 59.80 |
| | | SEA | 70.22 | 42.76 | 69.11 | 18.81 | 45.99 | 67.88 | 36.73 | 59.78 | 61.71 | 65.28 | 55.77 | 81.20 | 65.80 |
| | zh | ENG | 72.37 | 53.26 | 66.22 | 68.15 | 57.28 | 66.60 | 67.36 | 65.51 | 55.55 | 59.82 | 52.98 | 81.00 | 69.00 |
| | | SEA | 77.91 | 56.20 | 73.44 | 72.80 | 58.49 | 68.33 | 72.92 | 68.73 | 71.55 | 65.37 | 55.21 | 84.00 | 65.60 |
| **Average** | | ENG | 76.03 | 60.08 | 70.36 | 58.59 | 63.85 | 74.33 | 59.20 | 67.53 | 61.50 | 71.85 | 66.50 | 84.51 | 74.08 |
| | | SEA | 80.43 | 62.47 | 75.20 | 53.49 | 62.13 | 76.03 | 60.41 | 66.74 | 66.03 | 74.51 | 68.33 | 87.02 | 75.25 |

Table 15: TCQ performance: results across datasets comparing English and SEA prompts. Scores are reported as raw WER and CER values without normalization; lower values indicate better performance.

| Data | Lang | Length (Max) | Prompt | MERaLiON2 10B | MERaLiON2 3B | SeaLLMs Audio 7B | Phi-4 multi-modal instruct | Kimi Audio | Voxtral mini | Qwen2 Audio 7B Instruct | Qwen 2.5 Omni 3B | Qwen 2.5 Omni 7B | gemma 3n E4B-it | gemma 3n E2B-it | Gemini 2.5 Flash | GPT 4o Audio |
|---|---|---|---|---|---|---|---|---|---|---|---|---|---|---|---|---|
| Model / Size | | | | 10B | 3B | 7B | 5.6B | 7B | 3B | 7B | 3B | 7B | 4B | 2B | - | - |
| SG Streets | en | 30 | ENG | 2.43 | 2.43 | 2.12 | 2.67 | 2.47 | 1.89 | 2.30 | 2.64 | 4.21 | 2.20 | 2.32 | 0.47 | 2.24 |
| | | | SEA | 2.43 | 2.43 | 2.12 | 2.67 | 2.47 | 1.89 | 2.30 | 2.64 | 4.21 | 2.20 | 2.32 | 0.47 | 2.24 |
| | | 60 | ENG | 3.44 | 3.47 | - | 2.10 | 3.36 | 2.12 | - | 2.29 | 3.35 | 1.91 | 1.94 | 0.76 | 2.49 |
| | | | SEA | 3.44 | 3.47 | - | 2.10 | 3.36 | 2.12 | - | 2.29 | 3.35 | 1.91 | 1.94 | 0.76 | 2.49 |
| ASR-SgpCCSC | zh | 30 | ENG | 4.78 | 5.44 | 6.54 | 5.48 | 9.94 | 6.84 | 9.08 | 3.91 | 3.69 | 8.84 | 9.55 | 1.57 | 5.17 |
| | | | SEA | 4.34 | 5.13 | 5.68 | 4.07 | 10.43 | 5.46 | 8.87 | 4.15 | 2.91 | 8.62 | 9.67 | 1.57 | 4.99 |
| | | 60 | ENG | 9.85 | 11.28 | - | 6.72 | 22.74 | 12.86 | - | 3.85 | 5.73 | 9.98 | 9.36 | 1.89 | 8.00 |
| | | | SEA | 9.50 | 10.25 | - | 4.03 | 20.95 | 8.16 | - | 3.77 | 3.71 | 9.08 | 8.83 | 1.89 | 9.21 |
| | | 120 | ENG | 20.93 | 21.15 | - | 18.92 | 41.07 | 27.07 | - | - | 18.59 | - | - | 3.10 | 8.68 |
| | | | SEA | 20.86 | 17.09 | - | 14.94 | 37.36 | 12.12 | - | - | 8.19 | - | - | 3.10 | 10.81 |
| | | 180 | ENG | 20.59 | - | - | - | 57.41 | 36.85 | - | - | - | - | - | 3.30 | 7.35 |
| | | | SEA | 21.81 | - | - | - | 56.92 | 16.92 | - | - | - | - | - | 3.30 | 9.04 |
| YODAS2 | en | 30 | ENG | 5.34 | 5.31 | 3.81 | 4.53 | 5.79 | 3.62 | 4.57 | 4.94 | 4.42 | 5.03 | 5.03 | 1.36 | 4.42 |
| | | | SEA | 5.34 | 5.31 | 3.81 | 4.53 | 5.79 | 3.62 | 4.57 | 4.94 | 4.42 | 5.03 | 5.03 | 1.36 | 4.42 |
| | | 60 | ENG | 10.98 | 10.74 | - | 6.39 | 11.76 | 5.48 | - | 7.01 | 4.86 | 5.83 | 5.96 | 1.29 | 6.37 |
| | | | SEA | 10.98 | 10.74 | - | 6.39 | 11.76 | 5.48 | - | 7.01 | 4.86 | 5.83 | 5.96 | 1.29 | 6.37 |
| | | 120 | ENG | 16.19 | 15.10 | - | 9.02 | 22.28 | 9.09 | - | - | 6.92 | - | - | 1.82 | 6.49 |
| | | | SEA | 16.19 | 15.10 | - | 9.02 | 22.28 | 9.09 | - | - | 6.92 | - | - | 1.82 | 6.49 |
| | | 180 | ENG | - | - | - | - | - | - | - | - | - | - | - | 1.75 | 6.72 |
| | | | SEA | - | - | - | - | - | - | - | - | - | - | - | 1.75 | 6.72 |
| | id | 30 | ENG | 3.82 | 3.70 | 3.42 | 21.41 | 22.32 | 2.71 | 4.05 | 3.76 | 3.23 | 3.78 | 3.81 | 0.64 | 3.18 |
| | | | SEA | 3.77 | 3.50 | 3.52 | 14.13 | 9.58 | 3.06 | 3.86 | 3.81 | 4.54 | 4.00 | 3.91 | 1.36 | 3.20 |
| | | 60 | ENG | 7.88 | 6.44 | - | 14.18 | 19.97 | 4.40 | - | 5.26 | 4.09 | 4.23 | 4.30 | 1.21 | 4.47 |
| | | | SEA | 7.63 | 6.00 | - | 9.43 | 8.76 | 4.75 | - | 5.53 | 6.53 | 4.24 | 4.44 | 1.79 | 5.23 |
| | | 120 | ENG | 13.07 | 9.45 | - | 13.92 | 16.49 | 7.27 | - | - | 6.09 | - | - | 1.38 | 4.88 |
| | | | SEA | 12.59 | 9.08 | - | 12.35 | 10.94 | 6.27 | - | - | 8.20 | - | - | 2.37 | 4.89 |
| | | 180 | ENG | 14.00 | - | - | - | 16.24 | 10.26 | - | - | - | - | - | 2.02 | 5.42 |
| | | | SEA | 13.86 | - | - | - | 9.26 | 7.29 | - | - | - | - | - | 2.41 | 4.94 |
| | th | 30 | ENG | 3.82 | 1.49 | 5.31 | 80.08 | 27.81 | 5.24 | 3.14 | 36.29 | 9.68 | 9.21 | 10.49 | 10.15 | 7.09 |
| | | | SEA | 3.75 | 1.57 | 6.33 | 40.13 | 16.43 | 6.93 | 2.47 | 17.47 | 12.56 | 9.85 | 13.22 | 8.17 | 8.11 |
| | | 60 | ENG | 6.05 | 1.77 | - | 51.66 | 37.37 | 8.21 | - | 36.06 | 11.39 | 9.79 | 9.80 | 9.32 | 10.31 |
| | | | SEA | 5.63 | 1.73 | - | 37.39 | 17.88 | 9.97 | - | 23.94 | 13.33 | 9.74 | 9.89 | 7.43 | 11.61 |
| | | 120 | ENG | 10.10 | 2.10 | - | 41.25 | 32.92 | 12.71 | - | - | 11.80 | - | - | 10.16 | 11.10 |
| | | | SEA | 8.87 | 1.86 | - | 37.47 | 19.98 | 14.27 | - | - | 14.77 | - | - | 8.02 | 11.84 |
| | | 180 | ENG | 10.52 | - | - | - | 60.78 | 14.44 | - | - | - | - | - | 12.21 | 12.98 |
| | | | SEA | 9.76 | - | - | - | 27.66 | 16.71 | - | - | - | - | - | 9.93 | 12.87 |
| | vi | 30 | ENG | 5.48 | 5.22 | 4.11 | 21.25 | 17.42 | 4.28 | 4.42 | 3.06 | 3.11 | 5.30 | 5.95 | 1.26 | 4.33 |
| | | | SEA | 5.46 | 4.92 | 4.32 | 14.75 | 19.24 | 4.72 | 4.68 | 3.50 | 5.37 | 5.46 | 6.21 | 1.39 | 4.29 |
| | | 60 | ENG | 10.60 | 7.46 | - | 12.86 | 15.43 | 7.54 | - | 3.80 | 3.78 | 5.79 | 5.85 | 1.12 | 5.16 |
| | | | SEA | 10.42 | 6.25 | - | 11.12 | 17.28 | 7.88 | - | 3.73 | 6.13 | 5.89 | 5.84 | 1.46 | 5.90 |
| | | 120 | ENG | 14.05 | 8.90 | - | 15.76 | 11.24 | 11.77 | - | - | 4.81 | - | - | 46.66 | 5.90 |
| | | | SEA | 13.95 | 7.34 | - | 19.65 | 23.72 | 11.90 | - | - | 6.45 | - | - | 2.01 | 5.72 |
| | | 180 | ENG | 12.22 | - | - | - | 17.52 | 16.12 | - | - | - | - | - | 1.77 | 5.31 |
| | | | SEA | 12.03 | - | - | - | 26.28 | 12.47 | - | - | - | - | - | 2.06 | 4.55 |
| | zh | 30 | ENG | 10.16 | 9.83 | 13.04 | 9.29 | 12.21 | 8.61 | 11.38 | 6.57 | 10.10 | 12.95 | 14.12 | 1.41 | 8.99 |
| | | | SEA | 8.46 | 9.40 | 11.19 | 8.49 | 12.23 | 6.91 | 11.00 | 5.31 | 7.93 | 12.94 | 13.58 | 2.98 | 10.74 |
| | | 60 | ENG | 17.60 | 16.31 | - | 10.41 | 22.87 | 13.54 | - | 6.15 | 12.89 | 14.16 | 15.26 | 2.86 | 10.92 |
| | | | SEA | 15.95 | 14.34 | - | 11.48 | 23.07 | 8.33 | - | 4.67 | 9.09 | 13.88 | 15.08 | 4.40 | 13.01 |
| | | 120 | ENG | 28.66 | 22.18 | - | 14.92 | 44.31 | 22.13 | - | - | 20.73 | - | - | 2.94 | 14.59 |
| | | | SEA | 25.32 | 19.94 | - | 11.47 | 43.27 | 10.74 | - | - | 6.88 | - | - | 3.73 | 15.04 |
| | | 180 | ENG | 31.04 | - | - | - | 61.92 | 33.00 | - | - | - | - | - | 3.51 | 13.76 |
| | | | SEA | 28.22 | - | - | - | 73.53 | 14.62 | - | - | - | - | - | 4.49 | 17.39 |
| | | 180 | ENG | 17.67 | - | - | - | 42.78 | 22.13 | - | - | - | - | - | 4.09 | 8.59 |
| | | | SEA | 17.14 | - | - | - | 38.73 | 13.60 | - | - | - | - | - | 3.99 | 9.25 |
| | | 120 | ENG | 14.72 | 11.27 | - | 16.25 | 24.04 | 12.86 | - | - | 9.85 | - | - | 9.44 | 7.38 |
| | | | SEA | 13.97 | 10.06 | - | 14.99 | 22.51 | 9.20 | - | - | 7.34 | - | - | 3.01 | 7.83 |
| | | 30 | ENG | 5.12 | 4.77 | 5.48 | 20.67 | 14.00 | 4.74 | 5.56 | 8.74 | 5.49 | 6.76 | 7.32 | 2.41 | 5.06 |
| | | | SEA | 4.79 | 4.61 | 5.28 | 12.68 | 10.88 | 4.66 | 5.39 | 5.97 | 5.99 | 6.87 | 7.70 | 2.47 | 5.43 |
| | | 60 | ENG | 9.48 | 8.21 | - | 14.90 | 19.07 | 7.74 | - | 9.20 | 6.58 | 7.38 | 7.49 | 2.64 | 6.82 |
| | | | SEA | 9.08 | 7.54 | - | 11.70 | 14.72 | 6.67 | - | 7.28 | 6.71 | 7.22 | 7.42 | 2.72 | 7.69 |

Table 16: TLoc performance: results across datasets comparing English and SEA prompts. Evaluation is based on the F1 score metric, where higher values indicate better performance.

| Model | | | | MERaLiON2 10B | MERaLiON2 3B | SeaLLMs Audio 7B | Phi-4 multi-modal instruct | Kimi Audio | Voxtral mini | Qwen2 Audio 7B Instruct | Qwen 2.5 Omni 3B | Qwen 2.5 Omni 7B | gemma 3n E4B-it | gemma 3n E2B-it | Gemini 2.5 Flash | GPT 4o Audio |
|---|---|---|---|---|---|---|---|---|---|---|---|---|---|---|---|---|
| | | Size | | 10B | 3B | 7B | 5.6B | 7B | 3B | 7B | 3B | 7B | 4B | 2B | - | - |
| Data | Lang | Length (Max) | Prompt | | | | | | | | | | | | | |
| ASR-SgpCCSC | zh | 30 | ENG | 11.10 | 10.85 | 8.45 | 19.90 | - | - | 19.10 | 13.10 | 31.21 | 7.86 | 6.44 | 11.89 | 22.19 |
| | | | SEA | 11.10 | 10.85 | 8.45 | 19.90 | - | - | 19.10 | 13.10 | 31.21 | 7.86 | 6.44 | 11.89 | 22.19 |
| | | 60 | ENG | 6.23 | 6.42 | - | 7.73 | - | - | - | 10.78 | 12.71 | 6.64 | 2.88 | 1.26 | 10.50 |
| | | | SEA | 6.23 | 6.42 | - | 7.73 | - | - | - | 10.78 | 12.71 | 6.64 | 2.88 | 1.26 | 10.50 |
| | | 120 | ENG | 4.18 | 2.59 | - | 4.66 | - | - | - | - | 7.84 | - | - | 4.55 | 6.14 |
| | | | SEA | 4.18 | 2.59 | - | 4.66 | - | - | - | - | 7.84 | - | - | 4.55 | 6.14 |
| SG Streets | en | 30 | ENG | 40.28 | 33.56 | 22.56 | 11.96 | 17.17 | 27.70 | 54.85 | 49.61 | 44.77 | 18.04 | 19.18 | 16.62 | 40.25 |
| | | | SEA | 40.28 | 33.56 | 22.56 | 11.96 | 17.17 | 27.70 | 54.85 | 49.61 | 44.77 | 18.04 | 19.18 | 16.62 | 40.25 |
| | | 60 | ENG | 23.75 | 15.16 | - | 5.30 | 11.13 | 17.90 | - | 29.78 | 38.05 | 7.36 | 12.87 | 17.90 | 37.94 |
| | | | SEA | 23.75 | 15.16 | - | 5.30 | 11.13 | 17.90 | - | 29.78 | 38.05 | 7.36 | 12.87 | 17.90 | 37.94 |
| YODAS2 | en | 30 | ENG | 22.99 | 20.00 | 9.92 | 12.70 | 14.78 | 15.46 | 41.62 | 34.01 | 36.27 | 16.49 | 16.35 | 5.88 | 24.30 |
| | | | SEA | 22.99 | 20.00 | 9.92 | 12.70 | 14.78 | 15.46 | 41.62 | 34.01 | 36.27 | 16.49 | 16.35 | 5.88 | 24.30 |
| | | 60 | ENG | 13.91 | 11.46 | - | 6.79 | 8.43 | 7.76 | - | 21.07 | 18.67 | 10.44 | 11.36 | 7.98 | 17.14 |
| | | | SEA | 13.91 | 11.46 | - | 6.79 | 8.43 | 7.76 | - | 21.07 | 18.67 | 10.44 | 11.36 | 7.98 | 17.14 |
| | | 120 | ENG | 8.28 | 6.12 | - | 3.47 | 2.72 | 2.62 | - | - | 10.78 | - | - | 2.99 | 7.65 |
| | | | SEA | 8.28 | 6.12 | - | 3.47 | 2.72 | 2.62 | - | - | 10.78 | - | - | 2.99 | 7.65 |
| | id | 30 | ENG | 27.44 | 24.58 | 16.41 | 14.02 | 17.06 | 20.90 | 38.26 | 40.91 | 45.27 | 15.55 | 12.60 | 11.83 | 28.48 |
| | | | SEA | 26.70 | 18.43 | 19.78 | 10.63 | 15.04 | 18.26 | 37.56 | 21.84 | 34.71 | 20.16 | 16.19 | 16.19 | 30.01 |
| | | 60 | ENG | 14.90 | 12.99 | - | 7.47 | 9.94 | 11.76 | - | 23.78 | 21.44 | 10.50 | 9.66 | 11.04 | 20.84 |
| | | | SEA | 14.82 | 10.85 | - | 7.66 | 8.15 | 10.77 | - | 12.86 | 0.00 | 14.53 | 11.59 | 15.57 | 22.20 |
| | | 120 | ENG | 10.13 | 8.58 | - | 4.96 | 5.05 | 5.47 | - | - | 16.59 | - | - | 10.01 | 15.04 |
| | | | SEA | 10.43 | 6.45 | - | 5.18 | 4.91 | 4.17 | - | - | 13.73 | - | - | 9.42 | 15.60 |
| | th | 30 | ENG | 17.79 | 17.42 | 8.94 | 10.48 | 12.97 | 14.24 | 16.47 | 25.52 | 37.62 | 12.45 | 9.48 | 12.83 | 26.66 |
| | | | SEA | 16.04 | 15.32 | 9.16 | 13.75 | 13.02 | 10.84 | 11.92 | 14.08 | 27.42 | 10.71 | 6.93 | 25.73 | 17.62 |
| | | 60 | ENG | 9.03 | 8.56 | - | 5.90 | 8.03 | 7.16 | - | 14.76 | 15.22 | 9.28 | 7.72 | 4.35 | 18.64 |
| | | | SEA | 9.92 | 5.08 | - | 7.48 | 5.87 | 6.73 | - | 9.77 | 17.20 | 7.36 | 5.62 | 16.38 | 15.51 |
| | | 120 | ENG | 5.65 | 4.13 | - | 3.04 | 4.08 | 3.78 | - | - | 11.17 | - | - | 8.16 | 11.31 |
| | | | SEA | 5.87 | 1.94 | - | 3.70 | 5.80 | 3.00 | - | - | 6.75 | - | - | 15.81 | 9.18 |
| | vi | 30 | ENG | 22.40 | 14.50 | 8.22 | 10.30 | 15.84 | 18.81 | 35.08 | 33.18 | 39.17 | 12.14 | 10.35 | 10.11 | 25.67 |
| | | | SEA | 24.76 | 13.49 | 14.04 | 14.63 | 8.16 | 15.44 | 28.60 | 31.59 | 40.73 | 12.23 | 10.82 | 8.70 | 28.80 |
| | | 60 | ENG | 12.18 | 11.54 | - | 4.29 | 9.76 | 8.86 | - | 17.98 | 24.73 | 9.35 | 9.14 | 6.35 | 11.98 |
| | | | SEA | 14.77 | 7.92 | - | 7.70 | 11.35 | 7.30 | - | 17.96 | 28.44 | 10.95 | 8.88 | 7.28 | 21.40 |
| | | 120 | ENG | 6.34 | 7.34 | - | 2.44 | 4.84 | 3.76 | - | - | 12.73 | - | - | 2.83 | 7.40 |
| | | | SEA | 6.88 | 6.76 | - | 3.11 | 1.20 | 4.42 | - | - | 20.39 | - | - | 3.40 | 12.68 |
| | zh | 30 | ENG | 13.56 | 10.80 | 6.49 | 11.41 | 9.13 | 9.97 | 27.69 | 17.10 | 15.84 | 10.19 | 8.32 | 12.43 | 17.77 |
| | | | SEA | 12.52 | 10.61 | 6.69 | 11.02 | 11.41 | 11.14 | 26.21 | 13.93 | 16.46 | 12.36 | 7.29 | 10.11 | 14.80 |
| | | 60 | ENG | 6.61 | 5.85 | - | 6.28 | 4.36 | 5.76 | - | 9.35 | 9.07 | 7.45 | 4.29 | 5.02 | 7.73 |
| | | | SEA | 6.80 | 7.91 | - | 5.62 | 5.80 | 5.61 | - | 8.50 | 11.54 | 6.95 | 7.15 | 4.13 | 4.64 |
| | | 120 | ENG | 3.85 | 2.08 | - | 3.26 | 1.81 | 2.93 | - | - | 8.81 | - | - | 4.89 | 3.64 |
| | | | SEA | 3.98 | 4.38 | - | 3.16 | 0.00 | 2.35 | - | - | 4.77 | - | - | 6.32 | 4.34 |

# H USE OF LARGE LANGUAGE MODELS (LLMS)

We used LLMs solely as assist tools for polish writing. Specifically, we employed ChatGPT to improve grammar, clarity, and concision; refine figure captions and headings; and assist with LaTeX formatting (tables, subfigures, and minor package conflicts).

The LLM did not generate research ideas, design tasks or experiments, curate datasets, determine analyses, compute results, or author technical claims. All methods, experiments, and conclusions originate from the authors; all numbers and figures were produced by our code and independently checked by the authors.

The authors take full responsibility for the content of this manuscript. LLMs are not eligible for authorship and are not listed as authors.