# OpenReview forum: "SEA-SpeechBench: A Large-Scale Multitask Benchmark for Speech Understanding Across Southeast Asia"
_ICLR.cc/2026/Conference — ICLR 2026 Conference Withdrawn Submission_

### Official Review · Reviewer_hwBz · 2025-10-30

**Soundness:** 3
**Presentation:** 2
**Contribution:** 3
**Rating:** 4
**Confidence:** 4

**Summary:**

This paper introduces a novel and highly significant large-scale, multitask benchmark for evaluating speech understanding capabilities across 11 Southeast Asian (SEA) languages. This work directly addresses the critical lack of non-English evaluation frameworks, as current benchmarks are heavily English-centric, leaving a significant linguistic region severely underrepresented.

The benchmark comprises over 97,000 samples and 597 hours of curated audio data, covering 9 distinct tasks categorized into Speech Processing (e.g., ASR, ST), Paralinguistic Analysis (e.g., Age Recognition), and Temporal Understanding (e.g., SQA, TLoc). The paper provides a crucial evaluation of leading open-source and proprietary audio-language models, revealing a massive performance gap compared to English-centric performance.

**Strengths:**

1.	The paper is comprehensive and well presented.
2.	The motivation behind the paper is strong, tackling a major gap in multilingual speech research by providing the first comprehensive evaluation standard for a diverse and complex linguistic region.
3.	The scale and the inclusion of 11 SEA languages demonstrate substantial effort.

**Weaknesses:**

Given the highly uneven data distribution, the unweighted results can lead to critical performance drops in the low-resource languages that the benchmark is specifically designed to aid.

**Questions:**

1.	How was the judge model (Gemma 3 27B) selected? Wouldn’t there be bias issues for Gemma and Gemini results?
2.	The current Speech Translation (ST) tasks appear to be focused on SEA languages to English. Were any direct cross-lingual ST tasks between two Southeast Asian languages (e.g., Thai-to-Vietnamese) considered or included in the benchmark?
3.	Why does Table 4 use unweighted macro-averages, given the varying data sizes and the possible presence of data gaps between languages and tasks?
4.	Why were the Temporal Understanding tasks only applied to 5 out of the 11 languages? Specifically, what was the limiting factor that prevented this?

---

### Official Review · Reviewer_myMM · 2025-10-31

**Soundness:** 1
**Presentation:** 3
**Contribution:** 2
**Rating:** 2
**Confidence:** 4

**Summary:**

The paper proposes SEASpeechBench, a benchmark for audio LLMs on Southeast Asian speech that covers 9 tasks, categorized as "speech processing" (content-related), paralinguistic, and temporal understanding. Overall results show that existing models have poor performance, especially on low-resource languages and temporal reasoning, emotion recognition, and speech translation tasks. Failure modes of existing models are also showcased, such as tending to over-produce content for temporal understanding, and sensitivity to prompts in different languages.

**Strengths:**

* The study evaluates audio language models on Southeast Asian (SEA) speech, bringing attention to underrepresented SEA languages in mainstream speech processing research.
* The results shed light on various failure modes of existing models, such as tending to over-produce content for temporal understanding, and sensitivity to prompts in different languages.

**Weaknesses:**

* While the benchmark focuses on Southeast Asian (SEA) languages, its task coverage overlooks several core aspects of SEA speech. For instance, code-switching speech recognition (prevalent in Singapore, the Philippines, and Malaysia) and translation between SEA languages are not included, despite both being highly representative of real-world linguistic practices in the region.
* In addition, the dataset selection does not accurately reflect the speech characteristics of SEA speakers. For example, the English and Mandarin ASR data are drawn from CommonVoice, while the Mandarin speech translation data come from FLEURS. Neither of which capture the accentual, prosodic, and lexical variations typical of Southeast Asian nationals.
* The language coverage also omits several major regional languages. Examples include Hokkien and Cantonese (widely spoken in Singapore and Malaysia), Cebuano and Ilocano (major languages in the Philippines), Khmu (common in Laos), Hmong (spoken across Vietnam, Laos, and Thailand), Tetum (in Indonesia and Timor-Leste), and Javanese and Sundanese (in Indonesia). These omissions risk widening the gap between dominant and underrepresented languages.
* More generally, the task coverage remains limited. Key spoken language understanding (SLU) tasks—such as intent classification and slot filling—are not considered, even though these tasks hold particular promise for languages without standardized writing systems (especially important for SEA speech), offering a path toward more inclusive speech understanding.
* [Minor] there exists severe imbalance for data on specific tasks. For example, only 5 languages for emotion recognition, with thai, mandarin and English consist the majority.

**Questions:**

* The data generation process for spoken question answering is quite vague. What types of questions does the task cover?
* The “speech processing” task category is overly broad and ambiguous. It would benefit from being disaggregated into more precise subcategories that distinguish between phonological, prosodic, and semantic speech tasks.
* While the work addresses important linguistic aspects, its current framing and contribution are not well aligned with ICLR’s focus. It would be better suited for domain-specific venues such as Interspeech or SLT.
* The manuscript would benefit substantially from (1) proposing an (clever) audio data curation pipeline that contributes new datasets or closes existing language-resource gaps, particularly for underrepresented SEA languages; (2) providing a more in-depth discussion and analysis of the unique linguistic characteristics of Southeast Asian languages—such as code-switching, tonal variation, and morphological simplicity—to highlight what makes them distinctive and challenging for speech models.

---

### Official Review · Reviewer_oxY9 · 2025-11-01

**Soundness:** 2
**Presentation:** 3
**Contribution:** 3
**Rating:** 4
**Confidence:** 4

**Summary:**

The paper presents SEA-SpeechBench, a new benchmark for speech understanding tasks in 11 Southeast Asian languages. The benchmark tests the abilities of speech language models (SLMs) across 3 axes: speech processing, paralinguistics, and temporal understanding. The authors evaluated several open-source and proprietary SLMs on the benchmark, showing that models struggle to perform on these languages, despite recent improvements in English. The authors find that prompting in low-resource languages instead of English degrades performance, showing that model capabilities do not necessarily reflect real-life use cases.

**Strengths:**

- The authors benchmark 13 SLMs on various tasks. The authors construct evaluation sets for certain tasks in new languages which were previously unseen in literature (such as speaker recognition and sqa for various SEA languages), which will be important in measuring future progress in the field.
- The benchmark results show a significant bias towards using English-prompts instead of the native language prompts, a valuable finding that shows these models are heavily biased and do not adequately serve the needs of the SEA population.

**Weaknesses:**

-  For all non-ASR, ST, and TLoc tasks, model performance is evaluated using Gemma-3 27B Instruct. I believe this introduces some risk in the reported metrics, since the authors are using a weaker model (Gemma) to judge the capabilities of stronger ones, a studied failure case of the LLM-as-a-judge technique [1]. This is my main concern. The authors elaborate on their choice of Gemma "chosen for its demonstrated reliability in canonicalizing free-form text responses across Southeast Asian languages", but do not provide the necessary references or results to justify it.

- Its difficult to understand how well a model performs on each language (which I believe is what most users of this benchmark will care about), since the authors only report fully aggregated metrics, dataset-specific metrics and first-place results. Why not have a figure (like figure 3) that includes the performance of each model on each language and task?

- How is the averaging done when reporting each task metric? Is it across all datasets and languages? Across all datasets first and then languages? This will significantly affect the reported results and rankings [2,3]. Furthermore, many of the tasks do not even include all languages in the benchmark, which would skew the results towards that of English and Chinese, which appear to be the two most common languages.

- Since many tasks do not include all 11 languages, the claim that the benchmark is for 11 SEA languages is weak. As mentioned above, English and Chinese appear to be the two most common languages. Unless some kind of filtering was done, many of the corpora used for these two languages mostly do not contain variations commonly spoken in SEA (for ex. FLEURS, common-voice, yodas). These issues in yodas will have the most impact, since its 83% of the used audio.

[1] Krumdick, Michael, et al. "No free labels: Limitations of llm-as-a-judge without human grounding." arXiv preprint arXiv:2503.05061 (2025).

[2] Shi, Jiatong, et al. "Ml-superb 2.0: Benchmarking multilingual speech models across modeling constraints, languages, and datasets." Interspeech 2024.

[3] Chen, William, et al. "The ML-SUPERB 2.0 challenge: Towards inclusive ASR benchmarking for all language varieties." Interspeech 2025.

**Questions:**

- I think the the use of 1/(1+WER) for ASR makes the scores un-intuitive. Why not just report raw WER/CER and use the inversion formula only when calculating the average like [1]?

[1] Chen, Sanyuan, et al. "Wavlm: Large-scale self-supervised pre-training for full stack speech processing." IEEE Journal of Selected Topics in Signal Processing 16.6 (2022): 1505-1518.

---

### Note · Authors · 2025-11-28

I have read and agree with the venue's withdrawal policy on behalf of myself and my co-authors.